# Dimensionality Reduction on Complex Vector Spaces for Euclidean Distance with Dynamic Weights

Simone Moretti [1] [*]   Paolo Pellizzoni [2] [1] [*]   Francesco Silvestri [1]

## Abstract

The weighted Euclidean norm $\|x\|_w$ of a vector $x \in \mathbb{R}^d$ with weights $w \in \mathbb{R}^d$ is the Euclidean norm where the contribution of each dimension is scaled by a given weight. Approaches to dimensionality reduction that satisfy the Johnson–Lindenstrauss (JL) lemma can be easily adapted to the weighted Euclidean distance if weights are known and fixed: it suffices to scale each dimension of the input vectors according to the weights, and then apply any standard approach. However, this is not the case when weights are unknown during the dimensionality reduction or might dynamically change. In this paper, we address this issue by providing a linear function that maps vectors into a smaller complex vector space and allows to retrieve a JL-like estimate for the weighted Euclidean distance once weights are revealed. Our results are based on the decomposition of the complex dimensionality reduction into several Rademacher chaos random variables, which are studied using novel concentration inequalities for sums of independent Rademacher chaoses.

## 1. Introduction

The weighted Euclidean distance between two vectors is a Euclidean distance where the contribution of each dimension is scaled by a given weight representing the relevance of the dimension. Specifically, given a vector $x \in \mathbb{R}^d$ and a weight vector $w \in \mathbb{R}^d$ (with $w_i \geq 0$ for all $i \in \{1, \dots d\}$), the *weighted norm* of $x$ with weights $w$ is defined as $\|x\|_w = \sqrt{\sum_{i=1}^{d} w_i^2 x_i^2}$. The *weighted Euclidean distance* between two vectors $x, y$ is then defined as $\|x - y\|_w$.

---
[*]Equal contribution [1]University of Padova, Italy [2]Max Planck Institute of Biochemistry, Germany. Correspondence to: Francesco Silvestri <francesco.silvestri@unipd.it>.

*Proceedings of the 42nd International Conference on Machine Learning*, Vancouver, Canada. PMLR 267, 2025. Copyright 2025 by the author(s).

The weighted Euclidean distance is a frequently used primitive in several learning tasks for improving output quality, as weights can capture the estimated relevance of a feature on the output (Niño-Adan et al., 2021). In the context of LLMs, attention scores are used to weight the relative importance of each component in a sequence relative to the other elements in that sequence (Vaswani et al., 2017), while weighted low-rank approximation is used to compress models by taking into account the importance of parameters (Hsu et al., 2022; Woodruff & Yasuda, 2024). In recommendation systems, weights can improve the accuracy of prediction since it can enhance the role of relevant products while reducing the impacts of irrelevant products (Wang et al., 2015; Gu et al., 2016; Yu et al., 2003). Text $k$-NN classifiers use weights for taking into account the distribution of the terms across classes estimated from training data (Moreo et al., 2020; Bhattacharya et al., 2017). Most algorithmic methods supporting these applications assume that weights are known and fixed. However, weights might change over time or be not known at preprocessing time, a setting that we hereinafter refer to as "dynamic weights". Examples include nearest neighbor methods (Indyk & Motwani, 1998) for classification or recommender systems (Baumgärtner et al., 2022), where the relative importance of features might depend on the type of query at hand. Weights might be estimated on highly dynamic datasets that require a frequent update of the parameters (e.g., (Banihashem et al., 2024)). Moreover, weights might be user dependent and it might be too expensive to construct indexes for each possible set of weights, as in the aforementioned case of recommendation systems. For this reason, recent works have been addressing dynamic settings, such as for weighted Euclidean distance for the near neighbor search problem (Lei et al., 2019; Hu & Li, 2021).

The goal of this paper is to investigate the impact of dynamically weighted distance in dimensionality reduction techniques. Dimensionality reduction methods map high-dimensional vectors into a space with lower dimensionality, while keeping some information on the original vectors; the low-dimension vectors reduce communication and storage costs and mitigate the curse of dimensionality in the running time. A well-known result on dimensionality reduction is provided by the *Johnson–Lindenstrauss (JL) lemma* (John-

son & Lindenstrauss, 1984): given a set $\mathcal{X}$ of $n$ vectors in $\mathbb{R}^d$, there exists a linear map $g(\cdot)$ that approximately maintains pairwise squared Euclidean distances; that is, for any $x, y \in \mathcal{X}$, we have that $|\, \|x - y\|_2^2 - \|g(x) - g(y)\|_2^2 \,| \leq \varepsilon \|x - y\|_2^2$.

The JL lemma holds even with weighted Euclidean distance if weights are known beforehand. Consider one of the several linear maps $g(x)$ satisfying the JL lemma, for instance $g(x) = Ax$ where $A$ is the Achlioptas' matrix (Achlioptas, 2003) (i.e., each element is a random value in $\{-1, 1\}$). If all input vectors in $\mathcal{X}$ are scaled according to the given weights $w$ as $x' = w \odot x$, with $\odot$ denoting component-wise multiplication, the map $g(x')$ gives $|\, \|x - y\|_w^2 - \|g(x') - g(y')\|_2^2 \,| \leq \varepsilon \|x - y\|_w^2$, where $x'$ is the scaled version of the input vector $x \in \mathcal{X}$. However, this method cannot be applied if weights are known only after the dimensionality reduction step, or if weights could dynamically change. Note that naively storing the original vectors and scaling them once weights are provided would defeat the purpose of using dimensionality reduction. Therefore, one would need to find a dimensionality reduction technique that can reduce vectors obliviously to the weights, and then retrieve (an estimate of) the weighted norm once the weights are available based solely on the reduced vectors.

In fact, we observe that there is a simple, but fundamentally flawed, solution to this problem. Indeed, it is enough to observe that the weighted norm is the scalar product between the vector $(x \odot x)$ and $(w \odot w)$. Then, applying the JL dimensionality reduction to the vector $(x \odot x)$ yields the reduced vector $g'(x) = A(x \odot x)$, where $A$ is a $n \times k$ random matrix with entries in $\{-1, 1\}$. Then recalling the results of JL for scalar products, e.g. as shown in (Kaban, 2015), we have that $g'(x) \cdot g'(w)$ is an estimator of $\|x\|_w^2$ that can be computed from the reduced vector $g'(x)$ once the weights are revealed. In particular taking $k = O(\varepsilon^{-2} \ln(\delta^{-1}))$ yields, with probability at least $1 - \delta$,

$$|g'(x) \cdot g'(w) - \|x\|_w^2| < \varepsilon \|x\|_4^2 \|w\|_4^2. \tag{1}$$

However, the function $g'(\cdot)$ is *not linear* and it cannot be used for estimating pairwise weighted distances in a set of vectors, severely limiting the applicability of the method.

Indeed, the prime application of the linearity of JL maps is reducing the time complexity of pairwise distance computations (Cunningham & Ghahramani, 2015), which has applications in clustering (Makarychev et al., 2019) and nearest neighbor search (Indyk & Motwani, 1998; Ghalib et al., 2020), e.g. for recommender systems and classification. Other examples include compressed sensing (Upadhyay, 2015), which relies on sparse linear measurements to recover an unknown signal. Finally, the linearity of JL has been exploited to reduce the computational complexity

of least square regression (Yang et al., 2015) and low-rank matrix approximation (Ghojogh et al., 2021; Cunningham & Ghahramani, 2015).

Dimensionality reduction under Euclidean distance with dynamic weights can be formalized by the following problem: 1) each vector $x \in \mathcal{X}$ is mapped into a smaller vector with a *linear* function $g(\cdot)$, obliviously of weights; 2) when weights $w$ are provided, a function $\rho(g(x), w)$ is applied to a compressed vector $g(x)$ to obtain an unbiased estimate of $\|x\|_w^2$. To solve the problem we have to provide function $g(x)$ to map vectors into the smaller space, and function $\rho(g(x), w)$ to recover the weighted norm. To the best of our knowledge, no previous works have addressed this problem.

### 1.1. Our results

In this paper, we provide the first solution to dimensionality reduction with dynamic weights. We map vectors in $\mathcal{X}$ into a low dimensional *complex* vector space with a *linear* map $g(x) : \mathbb{R}^d \to \mathbb{C}^k$, for a suitable value $k > 0$. The map is done before the weights are revealed and it is hence independent of weights. More specifically, we let $g(x) = Ax/\sqrt{k}$ where matrix $A$ is a complex $k \times d$ random matrix, where each entry is an independently and identically distributed random variable over $\{+1, -1, +i, -i\}$, where $i$ is the imaginary unit (i.e., $i^2 = -1$). Once the weight vector $w$ is known, we apply a suitable function to the low-dimensional vectors in $\{g(x) : x \in \mathcal{X}\}$ that allows us to estimate the squared weighted Euclidean norms and the pairwise distances for any pair of vectors in $\mathcal{X}$. Let $h(g(x), w) = g(x) \odot g(x) \odot ((A \odot A)(w \odot w))$ with $\odot$ denoting the element-wise multiplication. Then, our unbiased estimate of $\|x\|_w^2$ is given by $\rho(g(x), w) = \text{Re}\big( \sum_{i=1}^{k} h(g(x), w)_i \big)$, where the subscript $\cdot_i$ denotes the $i$-th entry of a vector, and $\text{Re}(\cdot)$ is the real part. Note that computing $\rho(\cdot, \cdot)$ does not require access to the original vector $x$.

In the paper, we will show that the expected value of $\rho(g(x), w)$ is $\|x\|_w^2$ and provide upper bounds on the error. The following theorem, which is the main result of our paper, states that $\rho(g(x), w)$ provides a bounded additive error to the weighted norm, akin to the ones obtained for dot products under random projections (Kaban, 2015). Moreover, when the vector and weights at hand are such that the quantity $\|x\|_2^2 \|w\|_4^2 / \|x\|_w^2$ is bounded by some term $\Delta$, one can obtain a multiplicative approximation guarantee as in the JL lemma.

**Theorem 1.1.** *Let $\varepsilon, \delta > 0$. Let $\Delta$ be a suitable parameter and $k \geq \Omega\left(\max\left\{\frac{\Delta^2 \ln(8/\delta)}{\varepsilon^2}, \frac{\Delta \ln(8/\delta)^2}{\varepsilon}\right\}\right)$. Then there exists a linear function $g(x) : \mathbb{R}^d \to \mathbb{C}^k$ and an estimator $\rho(g(x), w) : \mathbb{C}^k \times \mathbb{R}^d \to \mathbb{R}$ such that for any given $x, w \in \mathbb{R}^d$, with probability at least $1 - \delta$,*

$$|\rho(g(x), w) - \|x\|_w^2| < \varepsilon \|w\|_4^2 \|x\|_2^2 / \Delta.$$

*In particular, if $\|x\|_2^2 \|w\|_4^2 / \|x\|_w^2 \le \Delta$, we get, with probability at least $1 - \delta$,*

$$|\rho(g(x), w) - \|x\|_w^2| < \varepsilon \|x\|_w^2.$$

We observe that the linearity of $g(\cdot)$ allows us to estimate pairwise weighted Euclidean distances using the compressed vectors, as $\rho(g(x) - g(y), w) = \rho(g(x - y), w)$ is a unbiased estimator of $\|x - y\|_w^2$.

We then extend, in Section 4.4, the previous result by showing that when the vectors $x$ and $w$ are near-uniformly distributed, we can lower the dimensionality of the reduced vectors to $\Theta\left(\max\left\{\left(\frac{\Delta}{\varepsilon}\right)\ln(8/\delta)^{1/2}, \left(\frac{\Delta}{\varepsilon}\right)^{2/3}\ln(8/\delta)^{4/3}\right\}\right)$, and still obtain the same guarantees. The approach uses a sparse block matrix consisting of $L$ submatrices of size $k \times d/L$, where each submatrix is generated as the aforementioned complex matrix $A$, for a suitable value of $L$.

The analysis of the aforementioned results leverages the decomposition of the complex dimensionality reduction into several Rademacher chaos random variables, i.e., sums of products of Rademacher random variables (see Section 2.3 for a formal definition). To do so, we develop a novel upper bound to tail probabilities for sums of independent Rademacher chaoses, which can be of independent interest.

Specifically, we have the following concentration result, which builds on a novel application of Bonami's hypercontractive inequality (Blei & Janson, 2004) to arbitrary Rademacher chaoses, and a recent concentration result for sums of independent sub-Weibull random variables (Kuchibhotla & Chakrabortty, 2022).

**Theorem 1.2.** *Let $S$ and $\{S_i\}_{i=1,\ldots,k}$ be $k + 1$ independent and identically distributed Rademacher chaoses of order $\gamma'$. Let $W(S)$ be the Euclidean norm of the coefficients in $S$. Then $\forall t > 0, \forall \gamma \ge 3$ such that $\gamma' \le \gamma$, there exist constants $c_1(\gamma), c_2(\gamma)$ depending only on $\gamma$ such that*

$$\Pr\left[\left|\sum_{i=1}^{k} S_i\right| > W(S)\left(c_1(\gamma)\sqrt{k}\sqrt{t} + c_2(\gamma)t^{\gamma/2}\right)\right] \le 2e^{-t}.$$

## 1.2. Related work

Recently, (Lei et al., 2019; Hu & Li, 2021) have studied the $r$-near neighbor search problem under weighted distance with weights provided at query time: given a set $S$ of $n$ vectors, construct a data structure that, for any query vector $x$ and weight vector $w$, returns a vector $y$ in $S$ such that $\|x - y\|_w \le r$. The papers leverage an asymmetric Locality Sensitive Hashing approach that uses a dimensionality expansion (from $\mathbb{R}^d$ to $\mathbb{R}^{2d}$) built with trigonometric functions. The result in (Lei et al., 2019) focuses on the weighted Euclidean distance, while the one in (Hu & Li, 2021) on the weighted Manhattan distance.

In (Har-Peled & Mahabadi, 2017), the authors have proposed a variant of the nearest neighbor search problem allowing for some coordinates of the dataset to be arbitrarily corrupted or unknown. The goal is to preprocess the input set such that, given a query point $q$, it is possible to efficiently find a point $x \in S$ such that the distance of the query $q$ to the point $x$ is minimized when removing the noisiest $k$ coordinates (i.e., coordinates are chosen so that deleting them from both vectors minimizes their distance). The paper provides a $(1 + \varepsilon)$-approximation to the optimal solution. We observe that in this work the coordinates to be removed are selected by the maximization procedure, and not specified in input.

Complex numbers have been used for approximately computing the permanent of a matrix (Chien et al., 2003), and in data streams to approximately count the number of occurrences of a given subgraph in a graph (Kane et al., 2012). A use of complex numbers similar to our case can be found in (Kane et al., 2011), where they estimate the $p$ norm in a streaming setting, with $0 < p < 2$.

The JL lemma has been proved to provide tight bounds (Larsen & Nelson, 2016; 2017), and has been used for several applications, including differential privacy (Blocki et al., 2012) and clustering (Makarychev et al., 2019).

Finally, Kaski et al. (2025) very recently addressed the problem of developing a generalization of JL for multi-linear dot products, which encompasses weighted norms. Their approach relies on non-independent matrices rather than complex numbers to achieve unbiasedness and only considers the case of general vectors. On the other hand, since we focus on the weighted norm, we develop an enhanced technique for near-uniform vectors, which yields better results.

## 1.3. Outline of the paper

In Section 2, we provide a formal background on the concepts we use throughout the paper. In Section 3, we derive the first tail bounds on sums of independent Rademacher chaoses. Our proof starts by generalizing the results of Bonami's hypercontractive inequality (Blei & Janson, 2004) to arbitrary Rademacher chaoses, which is then used to obtain an expression for the Orlicz norm of Rademacher chaoses. Using recent results based on such Orlicz norms (Kuchibhotla & Chakrabortty, 2022), we then prove the concentration result. In Section 4, we prove our main result (Theorem 1.1) and an improvement (Theorem 4.6) under the hypothesis of near-uniform vectors. Our proof is achieved by decomposing the estimator $\rho(g(x), w)$ into Rademacher chaoses and applying our novel concentration bounds.

All the formal proofs of lemmas and theorems, if not present in the main paper, are reported in the appendix.

## 2. Preliminaries

We now provide an overview of the concepts and techniques used to prove our results.

### 2.1. Basic concepts

Given a value $p > 0$, the *p-norm* of a vector $x \in \mathbb{R}^d$ is defined as $\|x\|_p = \left(\sum_{i=1}^d |x_i|^p\right)^{1/p}$. Given a *weight vector* $w$ in $\mathbb{R}^d_+$, the *weighted p-norm* of $x$ is $\|x\|_{w,p} = \left(\sum_{i=1}^d w_i^p |x_i|^p\right)^{1/p}$. When $p = 2$, we get the Euclidean norm and we simply denote the weighted norms of $x$ as $\|x\|_w$.

We state some commonly known inequalities that will be used in the subsequent sections.

**Lemma 2.1** (Cauchy-Schwarz ineq.)**.** *For any* $x, y \in \mathbb{R}^d$, $\left(\sum_{i=1}^d x_i y_i\right)^2 \leq \left(\sum_{i=1}^d x_i^2\right)\left(\sum_{i=1}^d y_i^2\right) = \|x\|_2^2 \|y\|_2^2$.

**Lemma 2.2** (Monotonicity of $p$-norm)**.** *For any* $x \in \mathbb{R}^d$ *and any* $0 < p < q < +\infty$, *we have* $\|x\|_q \leq \|x\|_p$.

**Lemma 2.3.** *For any* $p, q \geq 1$ *and any* $x, y \in \mathbb{R}^d$, *we have* $\sum_{i=1}^d |x_i|^p |y_i|^q \leq \|x\|_2^p \|y\|_2^q$.

### 2.2. Johnson-Lindenstrauss (JL) lemma

Let $\mathcal{X} = \{x_1, \ldots x_n\}$ be a set of $n$ vectors in $d > 0$ dimensions. For any $\varepsilon > 0$, the Johnson-Lindenstrauss lemma (Johnson & Lindenstrauss, 1984) claims that, if $k = \Omega\left(\varepsilon^{-2} \log n\right)$, there exists a linear function $g : \mathbb{R}^d \to \mathbb{R}^k$ that maintains pairwise distances up to a multiplicative error $\varepsilon$ with high probability. That is, for any $x, y \in \mathcal{X}$, we have

$$(1 - \varepsilon) \|x - y\|_2^2 \leq \|g(x) - g(y)\|_2^2 \leq (1 + \varepsilon) \|x - y\|_2^2.$$

Equivalently, the JL lemma claims the existence of a linear map that approximately maintains Euclidean norms. There are several constructions for $g(\cdot)$, for instance, those in (Achlioptas, 2003). Let $g(x) = Ax$ for a suitable $k \cdot d$ matrix; if $k = \Theta\left(\varepsilon^{-2} \log n\right)$ and each entry of $A$ is an independent and even distributed Rademacher random variable $\{-1, 1\}$, then $A$ satisfies the JL lemma with high probability. The property holds even if an entry of $A$ is 0 with probability 2/3 and in $\{-1, 1\}$ otherwise.

### 2.3. Rademacher chaos

We introduce here some results on Rademacher chaos. Let $I_{\gamma,n} \subseteq \{1, \ldots n\}^\gamma$, and let $I'_{\gamma,n} = \{(i_1, \ldots, i_\gamma) : i_j \in \{1, \ldots n\}, i_j \neq i_k \text{ if } j \neq k\}$ be a set of ordered sequences of $\gamma$ distinct values from $\{1, \ldots, n\}$; when $n$ is clear from the context, we drop the subscript $n$.

Let $X_1, \ldots X_n$ be $n$ independent Rademacher random variables. Then, a *Rademacher chaos* $S$ of order $\gamma$, with coeffi-

cients $a_{(i_1, \ldots i_\gamma)} \in \mathbb{R}$, is defined as:

$$S = \sum_{(i_1, \ldots i_\gamma) \in I'_{\gamma,n}} a_{(i_1, \ldots i_\gamma)} X_{i_1} \ldots X_{i_\gamma}.$$

The *q-norm* of the random variable $S$ is defined as $\|S\|_q = \mathbb{E}\left[|S|^q\right]^{1/q}$. We define a function $W$ on the set of all Rademacher chaoses as $W(S) = \sqrt{\sum_{i \in I'_{\gamma,n}} a^2_{(i_1, \ldots, i_\gamma)}}$, i.e. the Euclidean norm of the coefficients.

Bonami's hypercontractive inequality (Blei & Janson, 2004) provides a bound for $\|S\|_q$ when the sum is taken over sequences $(i_1, \ldots, i_\gamma)$ such that $i_1 < i_2 < \cdots < i_\gamma$, possibly after a renaming of the indexes.

**Theorem 2.4** ((Blei & Janson, 2004))**.** *Let* $S$ *be a Rademacher chaos of the form* $S = \sum_{1 \leq i_1 < \cdots < i_\gamma \leq n} a_{(i_1, \ldots i_\gamma)} X_{i_1} \ldots X_{i_\gamma}$. *Then,* $\|S\|_q \leq (q-1)^{\gamma/2} \|S\|_2 = (q-1)^{\gamma/2} W(S)$.

Note that in general, for Rademacher chaoses with sums over $I'_{\gamma,n}$, Bonami's hypercontractive inequality does *not* hold. In Section 3.1 we show how to generalize the result to arbitrary Rademacher chaoses.

### 2.4. Orlicz norms and sub-Weibull random variables

We introduce some results on the concentration of sums of independent heavy-tailed distributions. We recall the definition of Orlicz norms for random variables (Kuchibhotla & Chakrabortty, 2022).

**Definition 2.5.** Let $g : [0, \infty) \to [0, \infty)$ be a non-decreasing function with $g(0) = 0$. The *g-Orlicz norm* of a real-valued random variable $X$ is $\|X\|_g = \inf\{\eta > 0 : \mathbb{E}[g(|X|/\eta)] \leq 1\}$.

Note that, despite the name, if $g$ is not convex, the function $\|\cdot\|_g$ is not a norm. However, the norm properties are not needed in the following derivations.

Two well-known spacial cases of $g$ are $\psi_2 = \exp(x^2) - 1$ and $\psi_1 = \exp(x) - 1$. When a random variable has a finite $\psi_2$-Orlicz norm it is called sub-Gaussian and when it has a finite $\psi_1$-Orlicz norm it is called sub-exponential. The following definition (Kuchibhotla & Chakrabortty, 2022) generalizes such random variables.

**Definition 2.6.** A random variable $X$ is said to be sub-Weibull of order $\alpha > 0$ if $\|X\|_{\psi_\alpha} < \infty$ where $\psi_\alpha(x) = \exp(x^\alpha) - 1$.

The following theorem, proved in (Kuchibhotla & Chakrabortty, 2022) (Theorem 3.1), states a concentration result for sums of independent sub-Weibull random variables. We report a bound for heavy-tailed ($\alpha < 1$) variables.

**Theorem 2.7** ((Kuchibhotla & Chakrabortty, 2022))**.** *Let* $X_1, \ldots, X_k$ *be i.i.d. mean zero random variables with*

$\|X_i\|_{\psi_\alpha} < \infty$ *for some $\alpha \in (0,1)$. Then $\forall t > 0$*

$$\Pr\left[\left|\sum_{i=1}^k X_i\right| \geq C_\alpha \|X_1\|_{\psi_\alpha}(\sqrt{k}\sqrt{t} + 2^{2/\alpha - 1/2}t^{1/\alpha})\right] \leq 2e^{-t}$$

*with $C_\alpha = 8\sqrt{2}e^{97/24}(2\pi)^{1/4}(e^{2/e}/\alpha)^{1/\alpha}$.*

# 3. Concentration of sums of Rademacher chaoses

In this section, we provide the first concentration results for sums of independent Rademacher chaoses.

## 3.1. Tail bounds and Orlicz norms of arbitrary Rademacher chaoses

We generalize the bound given by Bonami's inequality for arbitrary Rademacher chaoses. Let $G_\gamma$ be the set of permutations over $\gamma$ elements, i.e. $G_\gamma = Sym(\{1, \dots, \gamma\})$. Then, $\forall \sigma \in G_\gamma$, let $I_{\gamma,n}^\sigma = \{(i_1, \dots, i_\gamma) : i_j \in \{1, \dots n\}, i_{\sigma(j)} < i_{\sigma(k)} \text{ if } j < k\}$. Then we can express

$$S = \sum_{(i_1, \dots i_\gamma) \in I'_{\gamma,n}} a_{(i_1, \dots i_\gamma)} X_{i_1} \dots X_{i_\gamma}$$

$$= \sum_{\sigma \in G_\gamma} \sum_{i \in I_{\gamma,n}^\sigma} a_{(i_1, \dots i_\gamma)} X_{i_1} \dots X_{i_\gamma} = \sum_{\sigma \in G_\gamma} S^\sigma$$

Intuitively, we can decompose the chaos $S$ into a collection of chaoses $S^\sigma$: in each $S^\sigma$, tuples can be sorted using permutation $\sigma$ and then Bonami's inequality can be applied. Indeed, Bonami's hypercontractive inequality (Blei & Janson, 2004) implies that, $\forall \sigma \, \forall q \geq 2$,

$$\|S^\sigma\|_q \leq (q-1)^{\gamma/2} \|S^\sigma\|_2$$

$$= (q-1)^{\gamma/2} \left(\sum_{(i_1, \dots i_\gamma) \in I_{\gamma,n}^\sigma} a_{(i_1, \dots i_\gamma)}^2\right)^{1/2}.$$

Using this decomposition, we obtain a bound on the $q$-norm of a general Rademacher chaos.

**Lemma 3.1.** *Let $S = \sum_{\sigma \in G_\gamma} S^\sigma$ be a Rademacher chaos of order $\gamma \geq 2$. Then, $\|S\|_q \leq (q-1)^{\gamma/2}\sqrt{\gamma!}W(S)$*

We then have the following concentration result.

**Lemma 3.2.** *Let $S$ be a Rademacher chaos of order $\gamma \geq 2$. Then for any $\gamma' \geq \gamma$ and $t > 0$, we have:*

$$\Pr\left[|S| > t\right] \leq e^2 \exp\left(-(e\sqrt{\gamma!}W(S))^{-2/\gamma'}t^{2/\gamma'}\right).$$

*Proof sketch.* We obtain the bound by applying Markov's inequality and Lemma 3.1 to obtain

$$\Pr\left[|S| > t\right] \leq \mathbb{E}\left[|S|^q\right]t^{-q} = \|S\|_q^q \, t^{-q},$$

and set $q$ appropriately. □

Finally, we can exploit the previous concentration result to provide an upper bound on the $\psi_\alpha$-Orlicz norm for a Rademacher chaos.

**Lemma 3.3.** *Let $S$ be a Rademacher chaos of order $\gamma \geq 2$. Let $\alpha \leq 2/\gamma$ and $\psi_\alpha = \exp(x^\alpha) - 1$. Then*

$$\|S\|_{\psi_\alpha} \leq e(e^2 + 1)^{1/\alpha}\sqrt{\gamma!}W(S).$$

*Proof.* From the definition of Orlicz norm we have $\|S\|_{\psi_\alpha} = \inf\{\eta > 0 : \mathbb{E}[\psi_\alpha(|S|/\eta)] \leq 1\}$. Then, applying Lemma 3.2 with $\gamma' = 2/\alpha \geq \gamma$, we get

$$\mathbb{E}[\psi_\alpha(|S|/\eta)] = \int_0^\infty \Pr[\psi_\alpha(|S|/\eta) > x]\,dx$$

$$= \int_0^\infty \Pr\left[|S| > \eta(\ln(x+1)^{1/\alpha})\right]dx$$

$$\leq e^2 \int_0^\infty \exp\left(-(e\sqrt{\gamma!}W(S))^{-\alpha}\eta^\alpha \ln(x+1)\right)dx$$

$$= e^2 \int_0^\infty (x+1)^{-\left(\frac{\eta}{e\sqrt{\gamma!}W(S)}\right)^\alpha}dx$$

$$= \frac{e^2}{\left(\frac{\eta}{e\sqrt{\gamma!}W(S)}\right)^\alpha - 1}.$$

Then, if $\eta \geq e(e^2 + 1)^{1/\alpha}\sqrt{\gamma!}W(S)$, we have that $\mathbb{E}[\psi(|S|/\eta)] \leq 1$. Therefore, we have that $\|S\|_{\psi_\alpha} \leq e(e^2 + 1)^{1/\alpha}\sqrt{\gamma!}W(S)$. □

## 3.2. Proof of Theorem 1.2

Having obtained a bound on the Orlicz norm for a Rademacher chaos, we can now prove the first of our main results, a concentration inequality for sums of independent Rademacher chaoses.

**Theorem 1.2.** *Let $S$ and $\{S_i\}_{i=1,\dots,k}$ be $k+1$ independent and identically distributed Rademacher chaoses of order $\gamma'$. Then $\forall t > 0$, $\forall \gamma \geq 3$ such that $\gamma' \leq \gamma$, there exist constants $c_1(\gamma), c_2(\gamma)$ depending only on $\gamma$ such that*

$$\Pr\left[\left|\sum_{i=1}^k S_i\right| > W(S)\left(c_1(\gamma)\sqrt{k}\sqrt{t} + c_2(\gamma)t^{\gamma/2}\right)\right] \leq 2e^{-t}.$$

*Proof.* By setting $\alpha = 2/\gamma \leq 2/\gamma'$ in Lemma 3.3, we get that $\|S\|_{\psi_{2/\gamma}} \leq e(e^2 + 1)^{\gamma/2}\sqrt{\gamma'!}W(S) \leq e(e^2 + 1)^{\gamma/2}\sqrt{\gamma!}W(S)$. Then, by defining $c_1(\gamma) = e(e^2 + 1)^{\gamma/2}\sqrt{\gamma!}C_{2/\gamma}$ and $c_2(\gamma) = e(e^2 + 1)^{\gamma/2}2^{\gamma-1/2}\sqrt{\gamma!}C_{2/\gamma}$, we get:

$$\Pr\left[\left|\sum_{i=1}^k S_i\right| > W(S)\left(c_1(\gamma)\sqrt{k}\sqrt{t} + c_2(\gamma)t^{\gamma/2}\right)\right]$$

$$\leq \Pr\left[\left|\sum_{i=1}^k S_i\right| > C_{2/\gamma}\|S\|_{\psi(2/\gamma)}\left(\sqrt{k}\sqrt{t} + 2^{\gamma-1/2}t^{\gamma/2}\right)\right].$$

Since $\mathbb{E}[S_i] = 0$ for any Rademacher chaos and by applying Theorem 2.7 with $\alpha = 2/\gamma$, we have that the right term of the previous inequality can be upper bounded as:

$$\Pr\left[\left|\sum_{i=1}^{k} S_i\right| > C_{2/\gamma} \|S\|_{\psi(2/\gamma)} \left(\sqrt{k}\sqrt{t} + 2^{\gamma-1/2} t^{\gamma/2}\right)\right] \leq 2e^{-t}.$$

The theorem then follows. $\qquad\square$

The above theorem provides the first tail bound for sums of independent Rademacher chaoses. The following corollary rephrases the tail bound by splitting it into a sub-Gaussian tail for small deviations and a heavy tail for large deviations.

**Corollary 3.4.** *Let $\{S_i\}_{i=1,\ldots,k}$ be a sequence of $k$ independent and identically distributed Rademacher chaoses of order $\gamma'$. Then $\forall \gamma \geq 3$ such that $\gamma' \leq \gamma$, there exist constants $c_1(\gamma), c_2(\gamma)$ depending only on $\gamma$ such that*

$$\Pr\left[\left|\sum_{i=1}^{k} S_i\right| > t\right] \leq \begin{cases} 2\exp\left(-\frac{1}{4c_1(\gamma)^2 W(S)^2} \frac{t^2}{k}\right) & \text{if } (\dagger) \\ 2\exp\left(-\left(\frac{t}{2c_2(\gamma)W(S)}\right)^{2/\gamma}\right) & \text{otherwise} \end{cases}$$

*with* $(\dagger)$ *the event:* $0 < t < 2\left(\frac{c_1(\gamma)^\gamma}{c_2(\gamma)}\right)^{\frac{1}{\gamma-1}} W(S)k^{\frac{\gamma}{2\gamma-2}}$

## 4. Complex dimensionality reduction

In this section, we analyze the properties of the dimensionality reduction onto the complex vector space described in the introduction. We recall that $A \in \mathbb{C}^{k \times d}$ denotes a random matrix where each entry in an independent and identically distributed random variable in $\{1, -1, i, -i\}$, and we define $g(x) = Ax/\sqrt{k}$ for any $x \in \mathbb{R}^d$. Let $h(g(x), w) = g(x) \odot g(x) \odot ((A \odot A)(w \odot w))$ and $\rho(g(x), w) = \text{Re}\left(\sum_{i=1}^{k} h(g(x), w)_i\right)$ be our estimate of $\|x\|_w^2$, where $\odot$ represents the element-wise multiplication. Although $\rho(g(x), w)$ is expressed as a function of $x$, it is computed from the reduced vector $g(x)$.

### 4.1. Results in expectation

We first show that the expected value of our estimator $\rho(g(x), w)$ corresponds to the weighted norm.

**Theorem 4.1.** *For any $x, w \in \mathbb{R}^d$, we have that $\mathbb{E}[\rho(g(x), w)] = \|x\|_w^2$.*

*Proof.* By the definition of $h(g(x), w)$, for any $i \in \{1, \ldots k\}$ we get:

$$h(g(x), w)_i = k^{-1} \left(\sum_{j=1}^{d} A_{i,j} x_j\right)^2 \left(\sum_{j=1}^{d} A_{i,j}^2 w_j^2\right)$$

$$= k^{-1} \sum_{(j_1, j_2, j_3) \in I_{3,d}} A_{i,j_1} A_{i,j_2} A_{i,j_3}^2 x_{j_1} x_{j_2} w_{j_3}^2.$$

Consider now the expectation of the term $\mathbb{E}\left[A_{i,j_1} A_{i,j_2} A_{i,j_3}^2\right]$ for a given value $i$. Since $\mathbb{E}[A_{i,j}] = \mathbb{E}\left[A_{i,j}^2\right] = \mathbb{E}\left[A_{i,j}^3\right] = 0$ for any $i, j$ and the terms in $A$ are independent, we have that $\mathbb{E}\left[A_{i,j_1} A_{i,j_2} A_{i,j_3}^2\right] = 1$ if $j_1 = j_2 = j_3$, and 0 otherwise. Then, we get:

$$\mathbb{E}[h(g(x), w)_i] = k^{-1} \sum_{j_1, j_2, j_3 \in I_{3,d}} \mathbb{E}\left[A_{i,j_1} A_{i,j_2} A_{i,j_3}^2\right] x_{j_1} x_{j_2} w_{j_3}^2$$

$$= k^{-1} \sum_{j=1}^{d} x_j^2 w_j^2 = \|x\|_w^2 / k.$$

The theorem then follows from the linearity of the sum and of the $\text{Re}(\cdot)$ function. $\qquad\square$

### 4.2. Results in high probability

We now prove, using as the main tool the newly developed Theorem 1.2, that the estimator for the weighted norm is concentrated around its mean with high probability. In particular, the concentration depends on $k$, the output space dimension. The concentration bounds then allow to choose $k$ depending on the desired accuracy of the estimate.

Since the results of Theorem 1.2 apply only to pairwise distinct indices, we split $\rho(g(x), w) = \sum_{i=1}^{k} \text{Re}\left(h_i(g(x), w)^2\right)$ into 4 sums, each over a set of pairwise distinct indices, and bound each term individually. Here we recall that $I_{\gamma,d} = \{1, \ldots, d\}^\gamma$ and $I'_{\gamma,d} = \{(j_1, \ldots, j_\gamma) \in I_{\gamma,d} : j_p \neq j_q \ \forall p, q \in \{1, \ldots \gamma\}, p \neq q\}$.

$$k \cdot \rho(g(x), w) = \sum_{i=1}^{k} \sum_{j_1, j_2, j_3 \in I_{3,d}} \text{Re}\left(A_{i,j_1} A_{i,j_2} A_{i,j_3}^2\right) x_{j_1} x_{j_2} w_{j_3}^2$$

$$= \sum_{i=1}^{k} \sum_{j_1, j_2, j_3 \in I'_{3,d}} \text{Re}\left(A_{i,j_1} A_{i,j_2} A_{i,j_3}^2\right) x_{j_1} x_{j_2} w_{j_3}^2$$

$$+ 2\sum_{i=1}^{k} \sum_{j_1, j_2 \in I'_{2,d}} \text{Re}\left(A_{i,j_1} A_{i,j_2}^3\right) x_{j_1} x_{j_2} w_{j_2}^2$$

$$+ \sum_{i=1}^{k} \sum_{j_1, j_2 \in I'_{2,d}} \text{Re}\left(A_{i,j_1}^2 A_{i,j_2}^2\right) x_{j_1}^2 w_{j_2}^2$$

$$+ \sum_{i=1}^{k} \sum_{j=1}^{d} w_j^2 x_j^2$$

$$= H_1 + H_2 + H_3 + k\|x\|_w^2.$$

**Remark.** It might be tempting to simply use a Khintchine-like inequality (Haagerup & Musat, 2007), such as in Kane et al. (2011, Lemma 4), to directly estimate $k \cdot \rho(g(x), w)$ using $A_{j_1} A_{j_2} A_{j_3}^2$ as the random variables and $x_{j_1} x_{j_2} w_{j_3}^2$ as their respective weights. This will however fail because the random variables are not independent. For this reason, we resort to Theorem 1.2 and Bonami's hypercontractive inequality to bound the tails of the Rademacher chaos.

Note that each $A_{i,j}$ can be written as $s(1 - r + i(1 + r))/2$, with $s$ and $r$ independent Rademacher random variables.

We then bound the tail probabilities of the individual terms $H_\ell$, $\ell = 1, 2, 3$ by expanding them into sums of Rademacher chaoses and using Theorem 1.2.

In particular, we decompose each $H_\ell$ as a sum of $k$ independent terms $H_{\ell,i}$, and we show that $W(H_{\ell,i})^2$, for any $i \in \{1, \ldots k\}$, can be upper bounded by $\|w\|_4^2 \|x\|_2^2$.

We report such bounds in the Appendix, and provide here a bound on $H = H_1 + H_2 + H_3$, which is obtained via a union bound over the three terms. In what follows, let $c_1$ and $c_2$ be the constants of Theorem 1.2 for $\gamma = 4$, and consequently $F(t) = c_1 \sqrt{k} \sqrt{t} + c_2 t^2$.

**Lemma 4.2.** *We have that* $\Pr\left[|H| > 4 \|w\|_4^2 \|x\|_2^2 F(t)\right] \leq 8e^{-t} \ \forall t > 0.$

We can now use the previous concentration result for $H$ to obtain a tail probability bound for the error $|\rho(g(x), w) - \|x\|_w^2|$ incurred by the estimator.

**Lemma 4.3.** *For any given $x, w \in \mathbb{R}^d$, we have*

$$\Pr\left[|\rho(g(x), w) - \|x\|_w^2| > t\right]$$
$$\leq \begin{cases} 8\exp\left(-k\frac{1}{(c_1 8 \|w\|_4^2 \|x\|_2^2)^2} t^2\right) & \text{if } (\dagger) \\ 8\exp\left(-\sqrt{k}\frac{1}{\sqrt{c_2 8 \|w\|_4^2 \|x\|_2^2}} \sqrt{t}\right) & \text{otherwise} \end{cases}$$

*with $(\dagger)$ the event:* $0 < t < c_1^{4/3} c_2^{-1/3} 8 \|w\|_4^2 \|x\|_2^2 k^{-1/3}.$

We then prove our main result Theorem 1.1. In particular, we obtain an additive error bound akin to the ones obtained for dot products under random projections (Kaban, 2015), which can be then refined to a multiplicative error by rescaling the accuracy parameter $\varepsilon$.

**Theorem 1.1.** *Let $\varepsilon, \delta > 0$. Let $\Delta$ be a suitable parameter and $k \geq \Omega\left(\max\left\{\frac{\Delta^2 \ln(8/\delta)}{\varepsilon^2}, \frac{\Delta \ln(8/\delta)^2}{\varepsilon}\right\}\right)$. Then there exists a linear function $g(x) : \mathbb{R}^d \to \mathbb{C}^k$ and an estimator $\rho(g(x), w) : \mathbb{C}^k \times \mathbb{R}^d \to \mathbb{R}$ such that for any given $x, w \in \mathbb{R}^d$, with probability at least $1 - \delta$,*

$$\left|\rho(g(x), w) - \|x\|_w^2\right| < \varepsilon \|w\|_4^2 \|x\|_2^2 / \Delta.$$

*In particular, if $\|x\|_2^2 \|w\|_4^2 / \|x\|_w^2 \leq \Delta$, we get, with probability at least $1 - \delta$,*

$$\left|\rho(g(x), w) - \|x\|_w^2\right| < \varepsilon \|x\|_w^2.$$

*Proof sketch.* We set $t = \varepsilon \|w\|_4^2 \|x\|_2^2 / \Delta$. For the Gaussian tail case of the bound described by Lemma 4.3, we take $k = c\Delta^2 \varepsilon^{-2} \ln(8/\delta)$.

Instead, for the long tail part of the bound described by Lemma 4.3, we take $k = c\Delta \frac{\ln(8/\delta)^2}{\varepsilon}$. Both cases yield $\Pr\left[|\rho(g(x), w) - \|x\|_w^2| > t\right] \leq \delta$.

As the failure probability decreases by $k$, taking $k \geq \max\left\{c_1^2 8^2 \frac{\Delta^2 \ln(8/\delta)}{\varepsilon^2}, c_2 8 \frac{\Delta \ln(8/\delta)^2}{\varepsilon}\right\}$ completes the proof. $\square$

Finally, we restate the above theorem to prove that the proposed reduction maintains pairwise distances in a set $S$ of $n$ vectors and for a set of weights $W$ with a bounded relative error. We remark that, thanks to the linearity of $g$, $\rho(g(x - y), w)$ can be computed from the reduced vector $g(x)$ and $g(y)$ as $\rho(g(x - y), w) = \rho(g(x) - g(y), w)$.

**Corollary 4.4.** *Let $\mathcal{X} = \{x_1, \ldots, x_n\}$, $x_i \in \mathbb{R}^d$ be a dataset of $n$ vectors. Let $W \subset \mathbb{R}^d$ be the set of the weights of interest, with $|W| = O(n)$. For any given $\varepsilon > 0$ and $k > c\max\left\{\Delta^2 \frac{\ln(8n)}{\varepsilon^2}, \frac{\Delta \ln(8n)^2}{\varepsilon}\right\}$ where $c$ is a universal constant, we have with high probability that, for any $x, y \in S$ and $w \in W$:*

$$\left|\rho(g(x - y), w) - \|x - y\|_w^2\right| < \varepsilon \|w\|_4^2 \|x - y\|_2^2 / \Delta.$$

*Proof.* It suffices to apply Theorem 1.1 with $x' = x - y$ and $\delta = 1/(|W|n^2)$. The claim follows by a union bound. $\square$

### 4.3. Discussion of results

We briefly discuss our results. We note that obtaining a multiplicative error guarantee on the estimate of the weighted norm $\|x\|_w^2$ using the dimensionality reduction given in Equation 1, which does not maintain linearity and therefore cannot be used to estimate weighted distances, requires the quantity $(\|x\|_4^2 \|w\|_4^2)/ \|x\|_w^2$ to be bounded. In particular, if $k = \Theta((\Delta/\varepsilon)^2)$ and $\max_{x,w}(\|x\|_4^2 \|w\|_4^2)/ \|x\|_w^2 \leq \Delta$, one obtains an $\varepsilon$ multiplicative error guarantee. This boundedness requirement is intuitively required by the fact that the vectors $x \odot x$ and $w \odot w$ could be orthogonal, and $\|x\|_w^2$ could therefore be 0. Similar results are obtained for estimating dot products under JL projections (Kaban, 2015).

According to Theorem 1.1, our novel complex dimensionality reduction technique, which instead guarantees linearity on $x$ and can therefore be applied to compute weighted distances (see Corollary 4.4), has a similar boundedness requirement. Indeed, if $k = \Theta((\Delta/\varepsilon)^2)$ and $\max_{x,w}(\|x\|_2^2 \|w\|_4^2)/ \|x\|_w^2 \leq \Delta$, one obtains a $\varepsilon$ multiplicative error guarantee. The dependence on the 2-norm instead of the 4-norm of $x$ slightly weakens the results. Indeed, we note that $\|x\|_2^2 \geq \|x\|_4^2$ (Lemma 2.2), and therefore our multiplicative error guarantee is worse than the non-linear method given by Equation 1. On one end of the spectrum, $w$ has a single non-zero entry, and $k = \Theta(\Delta^2)$

has to be $\Theta(\max_x\{\|x\|_2^4 / \min_i |x_i|^4\})$ to guarantee an $\varepsilon$-approximation. At the other end of the spectrum, $w$ is near-uniform (e.g., all the entries are within a factor $\zeta$), and $k = \Theta(\Delta^2)$ has to be $\Theta(\max_w \|w\|_4^4) = \Theta(d)$ to guarantee an $\varepsilon$-approximation. Nonetheless, for sparse weights such that the weighted norm is nonzero, the method yields a valid dimensionality reduction technique.

We observe that in the case where the weights $w$ are close to 1 (i.e., a small re-weighting of all the dimensions), the results of Theorem 1.1 yield no dimensionality reduction. We address this particular case in the following section.

### 4.4. Improved results for near-uniform vectors

We now deal with the case where the entries of both the $x$'s and $w$'s are near-uniform.

A uniform vector is a vector where all entries are the same. We define a (block) near-uniform vector as a relaxation of this condition, where splitting the vector into contiguous parts results in sub-vectors with roughly the same norm. More formally, let $1 \leq L \leq d$, and let $d_1, \ldots, d_L$ such that $\sum_\ell d_l = d$. Let then $x^{(\ell)} \in \mathbb{R}^{d_\ell}$, $\ell = 1, \ldots, L$ be such that $x = (x^{(1)}\|\ldots\|x^{(L)})$, with $\|$ denoting concatenation. The same holds for $w$. We suppose that $\left\|x^{(\ell)}\right\|_2^2 \leq \zeta \|x\|_2^2 /L$ and $\left\|w^{(\ell)}\right\|_4^4 \leq \zeta \|w\|_4^4 /L$, which holds, e.g., if all the entries are one within a factor $\zeta$ from any other one.

We will apply the dimensionality reduction to each of the sliced vectors individually, and then combine the estimates to obtain the final estimate for the weighted norm. We then have that our reduced vector is stored as $\bar{g}(x) = (g^{(\ell)}(x^{(\ell)}) : \ell = 1, \ldots, L)$, with each $g^{(\ell)}$ a function $g$, as in Section 4, defined with output dimension $k$ and pairwise independent matrices $A^{(\ell)}$. Each of such $L$ matrices is a complex $k \times d_\ell$ random matrix, where each entry is an independently and identically distributed random variable over $\{+1, -1, +i, -i\}$. Therefore, we have that $\bar{g}(x)$ can be stored with $k \cdot L$ complex numbers, i.e., $\bar{g}(x) \in \mathbb{C}^{k \cdot L}$. We observe that $\bar{g}(x)$ consists of a matrix-vector multiplication with a sparse block matrix $A$, made of $\ell$ submatrices of size $k \times d/\ell$ and where each submatrix is generated as the complex matrix described in the previous sections.

Let $\bar{\rho}(\bar{g}(x), w) = \sum_{\ell=1}^{L} \rho(g^{(\ell)}(x^{(\ell)}), w^{(\ell)})$, with $\rho$ defined as in Section 4. We can rewrite $\bar{\rho}$ as follows:

$$\bar{\rho}(\bar{g}(x), w) = \sum_{\ell=1}^{L} \rho(g^{(\ell)}(x^{(\ell)}), w^{(\ell)})$$

$$= \sum_{\ell=1}^{L} \sum_{i=1}^{k} \sum_{j_1, j_2, j_3 \in I_{3,d_\ell}} \mathrm{Re}\left(A_{i,j_1}^{(\ell)} A_{i,j_2}^{(\ell)} (A^{(\ell)})_{i,j_3}^2\right) \frac{x_{j_1}^{(\ell)} x_{j_2}^{(\ell)} (w_{j_3}^{(\ell)})^2}{k}$$

$$= H + \sum_{\ell=1}^{L} \left\|x^{(\ell)}\right\|_w^2 = H + \|x\|_w^2 .$$

We can then factorize $H$, which includes a sum over $I_{3,d}$ into three sums of Rademacher chaoses $H_1, H_2$ and $H_3$, as done in Section 4. Here, the main difference is that the sum that composes $H_1$ is over $k \cdot L$ independent chaoses $H_{1,\ell,i}$. These are not all identically distributed, but rather divided into $L$ blocks of $k$ identically distributed chaoses each. The same holds for $H_2$ and $H_3$.

In particular, we obtain that for chaoses in the $\ell$-th block, they satisfy $W(H_{1,\ell,i}) \leq \|x_\ell\|_2^2 \|w_\ell\|_4^4 /k \leq \zeta^{3/2} \|w\|_4^2 \|x\|_2^2 /(kL^{3/2})$.

Then, by following a derivation akin to the one in the previous section, which is reported in the appendix for the sake of brevity, we obtain the following result.

**Lemma 4.5.** *For any given $x, w \in \mathbb{R}^d$, we have $\forall t > 0$*

$$\Pr\left[|\bar{\rho}(\bar{g}(x), w) - \|x\|_w^2 | > t\right]$$

$$\leq \max \begin{cases} 8\exp\left(-kc_1 \dfrac{L^3}{\left(\|w\|_4^2 \|x\|_2^2\right)^2} t^2\right) \\ 8\exp\left(-\sqrt{k}c_2 \dfrac{L^{3/4}}{(\|w\|_4^2 \|x\|_2^2)^{1/2}} \sqrt{t}\right) \end{cases}$$

*with $c_1$ and $c_2$ appropriate constants.*

In turn, setting $k$ and $L$ carefully, we obtain a result akin to Theorem 1.1.

**Theorem 4.6.** *Let $\varepsilon, \delta > 0$. Let $c$ be a suitable universal constant. Let $L$ and $k$ be positive integers such that $k \geq c\max\left\{\dfrac{\Delta^2 \ln(8/\delta)}{L^2\varepsilon^2}, \dfrac{\Delta \ln(8/\delta)^2}{L^{3/2}\varepsilon}\right\}$. Then there exists a linear function $g(x) : \mathbb{R}^d \to \mathbb{C}^{k \cdot L}$ and an estimator $\rho(g(x), w) : \mathbb{C}^{k \cdot L} \times \mathbb{R}^d \to \mathbb{R}$ such that for any given $x, w \in \mathbb{R}^d$, with probability at least $1 - \delta$,*

$$\left|\rho(g(x), w) - \|x\|_w^2\right| < \varepsilon \|x\|_2^2 \|w\|_4^2 /\Delta.$$

*In particular, if $\|x\|_2^2 \|w\|_4^2 / \|x\|_w^2 \leq \Delta$, we get, with probability at least $1 - \delta$,*

$$(1 - \varepsilon) \|x\|_w^2 < \rho(g(x), w) < (1 + \varepsilon) \|x\|_w^2 .$$

We remark that, when the partition $d_1, \ldots, d_L$ is fixed for all vectors $x \in \mathcal{X}$, the function $\bar{g}$ is still linear, the dimensionality reduction maintains pairwise distances in a set of vectors $\mathcal{X}$.

### 4.5. Discussion of results for near-uniform vectors

In the case of near-uniform vectors, the results of Theorem 4.6 provide a significant improvement over the results of Theorem 1.1. Indeed, the balanced distribution across the entries of the vectors allows for more efficient dimensionality reduction via independent projections on the partitioned components of the vectors.

In particular, the output dimension $k \cdot L$ is decreasing in $L$. Therefore, if the vectors $x$ and $w$ are near-uniform for

large values of $L$, it is beneficial to set $k$ to be as small as possible, i.e., setting $k = 1$, and

$$L \geq c \max \left\{ (\Delta/\varepsilon) \ln (8/\delta)^{1/2}, (\Delta/\varepsilon)^{2/3} \ln (8/\delta)^{4/3} \right\}.$$

In this case, the dependency of the output dimension on $\Delta$ is lowered from up to $\Theta(\Delta^2)$ of the original technique, as discussed in Section 4.3, to only $\Theta(\Delta)$. In fact, we suppose that there exists some $\zeta > 1$ such that all entries of $w$ lie in the interval $[1/\sqrt{\zeta}, \sqrt{\zeta}]$ (i.e. no two entries of $x$ can be weighted arbitrarily differently). Then, we have that $\|w\|_4^2 \leq \zeta \sqrt{d}$ and $\|x\|_w^2 \geq 1/\zeta \|x\|_2^2$. Thus, setting $\Delta = \zeta^2 \sqrt{d} = \Theta(d^{1/2})$ ensures that $\|x\|_2^2 \|w\|_4^2 / \|x\|_w^2 \leq \Delta$. This means that to obtain a multiplicative $\varepsilon$-approximation to $\|x\|_w^2$, the dimension $k \cdot L = \Theta(\Delta)$ of the reduced vectors has to be only $\Theta(d^{1/2})$.

However, the technique to partition the vectors into $L$ smaller independent vectors guarantees a reduction in the dimensionality of the reduced vector only if the input vectors are guaranteed to be near-uniform. This is indeed a reasonable assumption on the weight vectors $w$, as one can expect to rescale the importance of each dimension just by a constant factor. On the other hand, to guarantee the near-uniformity condition on the input vectors $x$'s, one must have some prior knowledge on the data-generating distribution. We observe that, by bounding the maximum entry of the input and weight vectors, a random permutation of the dimension can provide a near-uniform distribution with high probability. Each entry is, e.g., uniformly distributed, one can get probabilistic guarantees via Bernstein's inequality.

Finally, we observe that the sparse block structure of the linear map allows the use of hardware accelerators for matrix multiplication (e.g., Google TPU, Nvidia TC) for further speeding up the computation. We leave the development of improvements to the dimensionality of the reduced vectors via partitioning strategies under weaker guarantees on the $x$'s to future work.

### 4.6. Experimental evaluation

In Appendix B, we report the results of a proof-of-concept experimental evaluation of our techniques. In particular, the experiments highlight the significant reduction of the variance of the estimator when using the sparse block matrices. Moreover, the experimental results showcase the applicability of the method on sparse vectors, as discussed in Section 4.3, as well as the poor quality of the estimate for near-uniform vectors for the method described in Theorem 1.2, suggesting that the probabilistic analysis of the method is tight. Finally, from the experiments, we indeed see a gap in the quality of the estimate between the linear and nonlinear method by Kaban (2015), as argued in Section 4.3. Interestingly, the sparse map with the decom-

position into $L$ sub-vectors almost closes the gap with the non-linear method, as argued in Section 4.5.

## 5. Discussion and Conclusions

In this paper, we provided the first dimensionality reduction techniques that are able to cope with dynamically-weighted Euclidean distances, as well as novel concentration inequalities for sums of independent Rademacher chaoses.

An interesting open question for future work is to provide a fast method for dimensionality reductions in the complex vector space similar to the Fast JL transform (Ailon & Chazelle, 2009). We furthermore conjecture that the complex map that we provide here admits a generalization to the estimation of arbitrary (weighted) $L_p$ norms. Indeed, let $h = \sum_{j_1, \ldots j_p} \prod_{k=1}^p A_{j_k} x_{j_k}$. If $A$ is a uniform on the $p$-th roots of 1, we have that $\mathbb{E}[\prod_{k=1}^p A_{j_k}] = 1$ if all the $j_k$'s are the same, and 0 otherwise. Therefore, $\mathbb{E}[h] = \|x\|_p^p$. However, the variance of the estimator grows with $p$, and novel approaches should be developed in order to reduce this growth.

We envision that an interesting research direction is the application of our techniques in downstream applications. For example, in machine learning, the weighted distance, phrased as the Mahalanobis distance with diagonal covariance matrix, is used as a time-efficient alternative to dynamic time warping (Prekopcsák & Lemire, 2010), and as a substitute for the Euclidean distance in RBF-like kernels (Abe, 2005; Kamada & Abe, 2006). These methods could then benefit from our techniques. We believe that our method might be of interest also in the weighted least squares problem (Golub & Van Loan, 2013, Chapter 5), which provides robust estimators in the presence of uneven reliability in the measurement (Fox, 2015). Here, given a data matrix $X \in \mathbb{R}^{n \times d}$, an observation vector $y \in \mathbb{R}^n$, and a weight vector $w \in \mathbb{R}^n$ (with $w_i \geq 0$ for all $i$), the goal is to find $\theta \in \mathbb{R}^d$ that minimizes the weighted residual norm $\|X\theta - y\|_w^2 = \sum_{i=1}^n w_i^2 (X_i^\top \theta - y_i)^2$. JL maps have been used to reduce the complexity of ordinary least squares (Yang et al., 2015), and we envision that our technique could be used in a similar fashion for weighted least squares under dynamic weights. Finally, we believe that our constructions can be of interest for privacy-preserving similarity search, as it might allow the release of datasets that allow users to detect if there are near points within the desired weighted norms without releasing the details of the vector, similar to what done in Blocki et al. (2012) for JL.

## Acknowledgments

This work was supported in part by MUR PRIN 20174LF3T8 AHeAD project, by MUR PNRR CN00000013 National Center for HPC, Big Data

and Quantum Computing, by Uni-Impresa Big-Mobility project, and by Marsden Fund (MFP-UOA2226).

## Impact Statement

This paper presents work whose goal is to advance the field of Machine Learning. There are many potential societal consequences of our work, none which we feel must be specifically highlighted here.

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

# Appendix

# A. Missing proofs

## A.1. Section 2

**Lemma 2.2.** For any $x \in \mathbb{R}^d$ and any $0 < p < q < +\infty$, we have $\|x\|_q \leq \|x\|_p$.

*Proof.* If $x = 0$ the claim holds trivially. Else, let $y_i = |x_i| / \|x\|_q \leq 1$. Then we have

$$\|x\|_p^p = \sum_{i=1}^d |x_i|^p = \sum_{i=1}^d |y_i|^p \|x\|_q^p \geq \sum_{i=1}^d |y_i|^q \|x\|_q^p = \|x\|_q^p \sum_{i=1}^d |x_i|^q / \|x\|_q^q = \|x\|_q^p.$$

$\square$

**Lemma 2.3.** For any $p, q \geq 1$ and any $x, y \in \mathbb{R}^d$, we have $\sum_{i=1}^d |x_i|^p |y_i|^q \leq \|x\|_2^p \|y\|_2^q$.

*Proof.* By applying Lemmas 2.1 and 2.2, we get:

$$\sum_{i=1}^d x_i^p y_i^q \leq \left( \sum_{i=1}^d |x_i|^{2p} \right)^{1/2} \left( \sum_{i=1}^d |y_i|^{2q} \right)^{1/2} \leq \|x\|_{2p}^p \|y\|_{2q}^q \leq \|x\|_2^p \|y\|_2^q.$$

$\square$

## A.2. Section 3

**Lemma 3.1.** Let $S = \sum_{\sigma \in G_\gamma} S^\sigma$ be a Rademacher chaos of order $\gamma \geq 2$. Then, $\|S\|_q \leq (q-1)^{\gamma/2} \sqrt{\gamma!} W(S)$.

*Proof.* Let $\overline{S}^\sigma = W(S^\sigma)^2 = \|S^\sigma\|_2^2 \ \forall \sigma$ and $\overline{S} = W(S)^2 = \sum_{\sigma \in G_\gamma} \overline{S}^\sigma$ We have that:

$$0 \leq \sum_{\sigma \in G_\gamma} \sum_{\rho \in G_\gamma} \left( \sqrt{\overline{S}^\sigma} - \sqrt{\overline{S}^\rho} \right)^2 = \sum_{\sigma \in G_\gamma} \sum_{\rho \in G_\gamma} \left( \overline{S}^\sigma + \overline{S}^\rho - 2\sqrt{\overline{S}^\sigma \overline{S}^\rho} \right) = 2\gamma! \sum_{\sigma \in G_\gamma} \overline{S}^\sigma - 2 \sum_{\sigma \in G_\gamma} \sum_{\rho \in G_\gamma} \sqrt{\overline{S}^\sigma \overline{S}^\rho} \leq$$

$$\leq 2\gamma! \sum_{\sigma \in G_\gamma} \overline{S}^\sigma - 2 \left( \sum_{\sigma \in G_\gamma} \sqrt{\overline{S}^\sigma} \right)^2.$$

This implies that $\sqrt{\gamma! \overline{S}} \geq \sum_{\sigma \in G_\gamma} \sqrt{\overline{S}^\sigma}$ and thus $\sqrt{\gamma!} W(S) \geq \sum_{\sigma \in G_\gamma} \|S^\sigma\|_2$. Then we can use the triangular inequality and Bonami's hypercontractive inequalities to prove the lemma:

$$\|S\|_q = \left\| \sum_{\sigma \in G_\gamma} S^\sigma \right\|_q \leq \sum_{\sigma \in G_\gamma} \|S^\sigma\|_q \leq (q-1)^{\gamma/2} \sum_{\sigma \in G_\gamma} \|S^\sigma\|_2 \leq (q-1)^{\gamma/2} \sqrt{\gamma!} W(S).$$

$\square$

**Lemma 3.2.** Let $S$ be a Rademacher chaos of order $\gamma \geq 2$. Then for any $\gamma' \geq \gamma$ and $t > 0$, we have:

$$\Pr[|S| > t] \leq e^2 \exp\left( -(e\sqrt{\gamma!} W(S))^{-2/\gamma'} t^{2/\gamma'} \right).$$

*Proof.* When $t \leq e\sqrt{\gamma!} W(S) 2^{\gamma'/2}$, the lemma trivially follows since $e^2 \exp\left( -(e\sqrt{\gamma!} W(S))^{-2/\gamma'} t^{2/\gamma'} \right) \geq 1$. Assume now that $t > e\sqrt{\gamma!} W(S) 2^{\gamma'/2}$. We apply Markov's inequality and lemma 3.1, obtaining

$$\Pr[|S| > t] \leq \mathbb{E}[|S|^q] t^{-q} = \|S\|_q^q t^{-q} \leq \left( \frac{q^{\gamma/2} \sqrt{\gamma!} W(S)}{t} \right)^q \leq \left( \frac{q^{\gamma'/2} \sqrt{\gamma!} W(S)}{t} \right)^q, \quad \forall q \geq 2.$$

Then, by setting $q = t^{2/\gamma'} (e\sqrt{\gamma! W(S)}))^{-2/\gamma'} \geq 2$, we have $\Pr[|S| > t] \leq \exp\left(-(e\sqrt{\gamma! W(S)})^{-2/\gamma'} t^{2/\gamma'}\right)$, from which the Lemma follows. $\qquad\square$

**Corollary 3.4.** Let $\{S_i\}_{i=1,\dots,k}$ be a sequence of $k$ independent and identically distributed Rademacher chaoses of order $\gamma'$. Then $\forall \gamma \geq 3$ such that $\gamma' \leq \gamma$, there exist constants $c_1(\gamma), c_2(\gamma)$ depending only on $\gamma$ such that

$$\Pr\left[\left|\sum_{i=1}^{k} S_i\right| > t\right] \leq \begin{cases} 2\exp\left(-\frac{1}{4c_1(\gamma)^2 W(S)^2}\frac{t^2}{k}\right) & \text{if } 0 < t < 2\left(\frac{c_1(\gamma)^\gamma}{c_2(\gamma)}\right)^{\frac{1}{\gamma-1}} W(S)k^{\frac{\gamma}{2\gamma-2}} \\ 2\exp\left(-\left(\frac{t}{2c_2(\gamma)W(S)}\right)^{2/\gamma}\right) & \text{otherwise.} \end{cases}$$

*Proof.* Let $\Delta = (c_1/c_2)^{\frac{2}{\gamma-1}} k^{\frac{1}{\gamma-1}}$ and let $\Delta' = 2\left(\frac{c_1^\gamma}{c_2}\right)^{\frac{1}{\gamma-1}} W(S)k^{\frac{\gamma}{2\gamma-2}}$. From Theorem 1.2 we have $\forall u > 0$

$$\Pr\left[\left|\sum_{i=1}^{k} S_i\right| > W(S)\left(c_1\sqrt{k}\sqrt{u} + c_2 u^{\gamma/2}\right)\right] \leq 2e^{-u}.$$

For $0 < u < \Delta$, we have that $c_1\sqrt{k}\sqrt{u} + c_2 u^{\gamma/2} \leq 2c_1\sqrt{k}\sqrt{u}$. Then,

$$\Pr\left[\left|\sum_{i=1}^{k} S_i\right| > 2c_1 W(S)\sqrt{k}\sqrt{u}\right] \leq \Pr\left[\left|\sum_{i=1}^{k} S_i\right| > W(S)\left(c_1\sqrt{k}\sqrt{u} + c_2 u^{\gamma/2}\right)\right] \leq 2e^{-u}.$$

By setting $u = t^2/\left(4c_1^2 W(S)^2 k\right)$, we have $u < \Delta$ when $t < \Delta'$. Then the above inequality gives the first part of the inequality.

For $u \geq \Delta$, we have that $c_1\sqrt{k}\sqrt{u} + c_2 u^{\gamma/2} \leq 2c_2 u^{\gamma/2}$. Then,

$$\Pr\left[\left|\sum_{i=1}^{k} S_i\right| > 2c_2 W(S)u^{\gamma/2}\right] \leq \Pr\left[\left|\sum_{i=1}^{k} S_i\right| > W(S)\left(c_1\sqrt{k}\sqrt{u} + c_2 u^{\gamma/2}\right)\right] \leq 2e^{-u}.$$

By setting $u = (t/(2c_2 W(S)))^{2/\gamma}$, we have $u \geq \Delta$ when $t \geq \Delta'$. Then the above inequality gives the second part of the inequality.

$\qquad\square$

### A.3. Section 4.2

We bound the tail probabilities of the individual terms $H_\ell$ by expanding them into sums of Rademacher chaoses and using Theorem 1.2. In what follows, let $F(t) = c_1\sqrt{k}\sqrt{t} + c_2 t^2$, with $c_1$ and $c_2$ the constants of Theorem 1.2 for $\gamma = 4$.

**Lemma A.1.** *We have that* $\Pr\left[|H_1| > \|w\|_4^2 \|x\|_2^2 F(t)\right] \leq 4e^{-t}. \ \forall t > 0$

*Proof.* By setting $\tau_{j_1,j_2,j_3} = x_{j_1} x_{j_2} w_{j_3}^2$, we rewrite $H_1$ as

$$H_1 = \sum_{i=1}^{k} \sum_{(j_1,j_2,j_3)\in I'_{3,d}} \text{Re}\left(A_{i,j_1} A_{i,j_2} A_{i,j_3}^2\right) \tau_{j_1,j_2,j_3}$$

$$= \sum_{i=1}^{k} \sum_{(j_1,j_2,j_3)\in I'_{3,d}} (s_{i,j_1} s_{i,j_2} r_{i,j_1} r_{i,j_3} + s_{i,j_1} s_{i,j_2} r_{i,j_2} r_{i,j_3})\tau_{j_1,j_2,j_3}/2$$

$$= \sum_{i=1}^{k} \sum_{(j_1,j_2,j_3)\in I'_{3,d}} s_{i,j_1} s_{i,j_2} r_{i,j_1} r_{i,j_3}\tau_{j_1,j_2,j_3}/2 + \sum_{i=1}^{k} \sum_{(j_1,j_2,j_3)\in I'_{3,d}} s_{i,j_1} s_{i,j_2} r_{i,j_2} r_{i,j_3}\tau_{j_1,j_2,j_3}/2$$

$$= \sum_{i=1}^{k} H_{1,1,i}/2 + \sum_{i=1}^{k} H_{1,2,i}/2 = H_{1,1}/2 + H_{1,2}/2.$$

Since $H_{1,1}$ and $H_{1,2}$ have the same distribution, we focus on the first one. There are three indexes but four independent random variables in each term of the sum in $H_{1,1}$, we perform an index transformation to get a chaos of order 4. Let $\hat{\tau} : I'_{4,2d} \to \mathbb{R}$ be

$$\hat{\tau}_{j_1,j_2,j'_1,j_3} = \begin{cases} \tau_{j_1,j_2,j_3-d} & \text{if } j'_1 = j_1 + d, j_1, j_2 \le d, j_3 > d, j_3 \ne j_2 + d \\ 0 & \text{otherwise.} \end{cases}$$

Let also $X_{i,:} = (s_{i,1}, \dots, s_{i,d}, r_{i,1}, \dots, r_{i,d})$. Note that the components of $X$ are independent. Then, we can rewrite $H_{1,1}$ as

$$H_{1,1} = \sum_{i=1}^{k} \sum_{(j_1,j_2,j'_1,j_3) \in I'_{4,2d}} X_{i,j_1} X_{i,j_2} X_{i,j'_1} X_{i,j_3} \hat{\tau}_{j_1,j_2,j'_1,j_3}.$$

We now upper bound $W(H_{1,1,i})^2$ for any $i \in \{1, \dots k\}$; standard arguments give

$$W(H_{1,1,i})^2 = \sum_{(j_1,j_2,j'_1,j_3) \in I'_{4,2d}} \hat{\tau}^2_{j_1,j_2,j'_1,j_3} = \sum_{(j_1,j_2,j_3) \in I'_{3,d}} \left(x_{j_1} x_{j_2} w^2_{j_3}\right)^2$$

$$= \sum_{(j_1,j_2,j_3) \in I'_{3,d}} x^2_{j_1} x^2_{j_2} w^4_{j_3} \le \|w\|^4_4 \|x\|^4_2.$$

The final inequality holds as:

$$\sum_{(j_1,j_2,j_3) \in I'_{3,d}} x^2_{j_1} x^2_{j_2} w^4_{j_3} \le \sum_{(j_1,j_2,j_3) \in I_{3,d}} |x^2_{j_1} x^2_{j_2} w^4_{j_3}|$$

$$\le \left(\sum_{j_1 \in I_{1,d}} |x_{j_1}|^2\right) \left(\sum_{j_2 \in I_{1,d}} |x_{j_2}|^2\right) \left(\sum_{j_3 \in I_{1,d}} |w_{j_3}|^4\right)$$

$$= \|w\|^4_4 \|x\|^4_2.$$

We can then apply Theorem 1.2 with $\gamma = 4$, from which follows that

$$\Pr\left[|H_{1,1}| > \|w\|^2_4 \|x\|^2_2 F(t)\right] \le \Pr\left[|H_{1,1}| > W(H_{1,1,i})F(t)\right] \le 2e^{-t}.$$

Since a similar bound holds for $H_{2,2}$, and since $2H_1 = H_{1,1} + H_{1,2}$, an union bound gives

$$\Pr\left[|H_1| > \|w\|^2_4 \|x\|^2_2 F(t)\right] = \Pr\left[|H_{1,1} + H_{1,2}| > 2 \|w\|^2_4 \|x\|^2_2 F(t)\right]$$

$$\le 2\Pr\left[|H_{1,1}| > \|w\|^2_4 \|x\|^2_2 F(t)\right] \le 4e^{-t}.$$

$\square$

**Lemma A.2.** *We have that* $\Pr\left[|H_2| > 2 \|w\|^2_4 \|x\|^2_2 F(t)\right] \le 2e^{-t}. \ \forall t > 0$

*Proof.* By setting $\tau_{j_1,j_2} = x_{j_1} x_{j_2} w^2_{j_2}$, we rewrite $H_2$ as

$$H_2 = 2 \sum_{i=1}^{k} \sum_{(j_1,j_2) \in I'_{2,d}} \text{Re}\left(A_{i,j_1} A^3_{i,j_2}\right) \tau_{j_1,j_2}$$

$$= 2 \sum_{i=1}^{k} \sum_{(j_1,j_2) \in I'_{2,d}} s_{j_1} s_{j_2}(1 + r_{j_1} r_{j_2})\tau_{j_1,j_2}/2$$

$$= \sum_{i=1}^{k} \sum_{(j_1,j_2) \in I'_{2,d}} s_{j_1} s_{j_2} \tau_{j_1,j_2} + \sum_{i=1}^{k} \sum_{(j_1,j_2) \in I'_{2,d}} s'_{j_1} s'_{j_2} \tau_{j_1,j_2}$$

where $s'_j = s_j r_j$ and has the same distribution of a Rademacher random variable.

Let $\hat{\tau} : I'_{2,2d} \to \mathbb{R}$ be

$$
\hat{\tau}_{j_1,j_2} = \begin{cases} \tau_{j_1,j_2} & \text{if } j_1 \leq d, j_2 \leq d \\ \tau_{j_1-d,j_2-d} & \text{if } j_1 > d, j_2 > d \\ 0 & \text{otherwise.} \end{cases}
$$

Let also $X_{i,:} = (s_{i,1}, \ldots, s_{i,d}, s'_{i,1}, \ldots, s'_{i,d}, )$. Note that the components of $X$ are independent.

Then, we can rewrite $H_2$ as

$$
H_2 = \sum_{i=1}^{k} \sum_{(j_1,j_2) \in I'_{2,2d}} X_{i,j_1} X_{i,j_2} \hat{\tau}_{j_1,j_2} = \sum_{i=1}^{k} H_{2,i}.
$$

Then, we get that:

$$
W(H_{2,i})^2 = \sum_{(j_1,j_2) \in I'_{2,2d}} \hat{\tau}_{j_1,j_2}^2 = 2 \sum_{(j_1,j_2) \in I'_{2,d}} \left( x_{j_1} x_{j_2} w_{j_2}^2 \right)^2 = 2 \sum_{(j_1,j_2) \in I'_{2,d}} x_{j_1}^2 x_{j_2}^2 w_{j_2}^4 \leq 4 \|w\|_4^4 \|x\|_2^4 .
$$

Since $H_2$ is a sum of independent Rademacher chaoses of order $2 \leq 4$, from Theorem 1.2 we have that

$$
\Pr \left[ |H_2| > 2 \|w\|_4^2 \|x\|_2^2 F(t) \right] \leq \Pr \left[ |H_2| > W(H_{2,i}) F(t) \right] \leq 2e^{-t},
$$

and the claim follows. □

**Lemma A.3.** *We have that* $\Pr \left[ |H_3| > \|w\|_4^2 \|x\|_2^2 F(t) \right] \leq 2e^{-t}. \ \forall t > 0$

*Proof.* By setting $\tau_{j_1,j_2} = x_{j_1}^2 w_{j_2}^2$, we rewrite $H_3$ as

$$
H_3 = \sum_{i=1}^{k} \sum_{(j_1,j_2) \in I'_{2,d}} \mathrm{Re}\left( A_{i,j_1}^2 A_{i,j_2}^2 \right) \tau_{j_1,j_2} = \sum_{i=1}^{k} \sum_{(j_1,j_2) \in I'_{2,d}} r_{j_1} r_{j_2} \tau_{j_1,j_2} = \sum_{i=1}^{k} H_{3,i}
$$

Then we get:

$$
W(H_{3,i})^2 = \sum_{(j_1,j_2) \in I'_{2,d}} x_{j_1}^4 w_{j_2}^4 \leq \|w\|_4^4 \|x\|_2^4 .
$$

Since $H_3$ is a sum of independent Rademacher chaoses of order $2 \leq 4$, from Theorem 1.2 we have that

$$
\Pr \left[ |H_3| > \|w\|_4^2 \|x\|_2^2 F(t) \right] \leq \Pr \left[ |H_3| > W(H_{3,i}) F(t) \right] \leq 2e^{-t},
$$

and the claim follows. □

We then provide a bound on the quantity $H = H_1 + H_2 + H_3$.

**Lemma 4.2.** We have that $\Pr \left[ |H| > 4 \|w\|_4^2 \|x\|_2^2 F(t) \right] \leq 8e^{-t}. \ \forall t > 0$

*Proof.*

$$
\Pr \left[ |H| > 4 \|w\|_4^2 \|x\|_2^2 F(t) \right] = \Pr \left[ |H_1 + H_2 + H_3| > 4 \|w\|_4^2 \|x\|_2^2 F(t) \right] \leq
$$
$$
\leq \Pr \left[ |H_1| > \|w\|_4^2 \|x\|_2^2 F(t) \right] + \Pr \left[ |H_2| > 2 \|w\|_4^2 \|x\|_2^2 F(t) \right] + \Pr \left[ |H_3| > \|w\|_4^2 \|x\|_2^2 F(t) \right]
$$
$$
\leq 4e^{-t} + 2e^{-t} + 2e^{-t} = 8e^{-t}.
$$

□

**Lemma 4.3.** For any given $x, w \in \mathbb{R}^d$, we have

$$\Pr\left[|\rho(g(x), w) - \|x\|_w^2| > t\right]$$

$$\leq \begin{cases} 8\exp\left(-k\dfrac{1}{(c_1 8\|w\|_4^2\|x\|_2^2)^2}t^2\right) & \text{if } (\dagger) \\ 8\exp\left(-\sqrt{k}\dfrac{1}{\sqrt{c_2 8\|w\|_4^2\|x\|_2^2}}\sqrt{t}\right) & \text{otherwise} \end{cases}$$

with $(\dagger)$ the event: $0 < t < c_1^{4/3}c_2^{-1/3}8\|w\|_4^2\|x\|_2^2 k^{-1/3}$.

*Proof.* We have $k\rho(g(x), w) - k\|x\|_w^2 = H_1 + H_2 + H_3 = H$, which in turn implies that

$$\Pr\left[|\rho(g(x), w) - \|x\|_w^2| > t\right] = \Pr[|H| > kt] = \Pr[|H| > u].$$

From Lemma 4.2, we have that

$$\Pr\left[|H| > 4\|w\|_4^2\|x\|_2^2\left(c_1\sqrt{k}\sqrt{v} + c_2 v^2\right)\right] \leq 8e^{-v}.$$

Then, for $v < (c_1/c_2)^{2/3}k^{1/3}$, we have that $2c_1\sqrt{k}\sqrt{v} \geq c_1\sqrt{k}\sqrt{v}+c_2 v^2$ and therefore $\Pr\left[|H| > 8\|w\|_4^2\|x\|_2^2 c_1\sqrt{k}\sqrt{v}\right] \leq 8e^{-v}$. Similarly, for $v \geq (c_1/c_2)^{2/3}k^{1/3}$, we have that $\Pr\left[|H| > 8\|w\|_4^2\|x\|_2^2 c_2 v^2\right] \leq 8e^{-v}$.

Then, setting $u = 8\|w\|_4^2\|x\|_2^2 c_1\sqrt{k}\sqrt{v}$ for the first bound and $u = 8\|w\|_4^2\|x\|_2^2 c_2 v^2$ for the second one yields

$$\Pr[|H| > u] \leq \begin{cases} 8\exp\left(-\dfrac{1}{(c_1 8\|w\|_4^2\|x\|_2^2)^2}\dfrac{u^2}{k}\right) & \text{if } 0 < u < c_1^{4/3}c_2^{-1/3}8\|w\|_4^2\|x\|_2^2 k^{2/3} \\ 8\exp\left(-\dfrac{1}{\sqrt{c_2 8\|w\|_4^2\|x\|_2^2}}\sqrt{u}\right) & \text{otherwise.} \end{cases}$$

Finally, making the substitution $u = kt$, we obtain the claim. $\qquad\square$

**Theorem 1.1.** *Let $\varepsilon, \delta > 0$. Let $\Delta$ be a suitable parameter and $k \geq \Omega\left(\max\left\{\frac{\Delta^2\ln(8/\delta)}{\varepsilon^2}, \frac{\Delta\ln(8/\delta)^2}{\varepsilon}\right\}\right)$. Then there exists a* linear *function $g(x) : \mathbb{R}^d \to \mathbb{C}^k$ and an estimator $\rho(g(x), w) : \mathbb{C}^k \times \mathbb{R}^d \to \mathbb{R}$ such that for any given $x, w \in \mathbb{R}^d$, with probability at least $1 - \delta$,*

$$|\rho(g(x), w) - \|x\|_w^2| < \varepsilon\|x\|_2^2\|w\|_4^2/\Delta.$$

*In particular, if $\|x\|_2^2\|w\|_4^2 / \|x\|_w^2 \leq \Delta$, we get, with probability at least $1 - \delta$,*

$$|\rho(g(x), w) - \|x\|_w^2| < \varepsilon\|x\|_w^2.$$

*Proof.* Assume that $\varepsilon\ln(8/\delta)/\Delta < c_1^2 c_2^{-1}8$, and set $k = c_1^2 8^2\Delta^2\varepsilon^{-2}\ln(8/\delta)$, and we get:

$$t = \varepsilon\|w\|_4^2\|x\|_2^2/\Delta < c_1^{4/3}c_2^{-1/3}8\|w\|_4^2\|x\|_2^2 k^{-1/3},$$

We are thus in the Gaussian tail case of the bound described by Lemma 4.3. Then we have

$$\Pr\left[|\rho(g(x), w) - \|x\|_w^2| > t\right] \leq 8\exp\left(-\frac{kt^2}{(c_1 8\|w\|_4^2\|x\|_2^2)^2}\right) \leq 8\exp\left(-c_1^2 8^2\varepsilon^{-2}\ln\left(\frac{8}{\delta}\right)\frac{\varepsilon^2}{c_1^2 8^2}\right) = \delta.$$

Let now assume $\varepsilon\ln(8/\delta)/\Delta \geq c_1^2 c_2^{-1}8$, and fix $k = c_2 8\Delta\frac{\ln(8/\delta)^2}{\varepsilon}$. In this case,

$$t = \varepsilon\|x\|_2^2\|w\|_4^2/\Delta \geq c_1^{4/3}c_2^{-1/3}8\|w\|_4^2\|x\|_2^2 k^{-1/3},$$

and we are in the long tail part of the bound described by Lemma 4.3. Then,

$$\Pr\left[\,|\rho(g(x),w) - \|x\|_w^2| > t\right] \le 8\exp\left(-\frac{\sqrt{kt}}{\sqrt{c_2 8\,\|w\|_4^2\,\|x\|_2^2}}\right) \le 8\exp\left(-c_1^2 8^2 \varepsilon^{-2}\ln\left(\frac{8}{\delta}\right)\frac{\varepsilon^2}{c_1^2 8^2}\right) = \delta.$$

As the failure probability decreases by $k$, taking $k \ge \max\left\{c_1^2 8^2\frac{\Delta^2\ln(8/\delta)}{\varepsilon^2},\; c_2 8\frac{\Delta\ln(8/\delta)^2}{\varepsilon}\right\}$ completes the proof. $\qquad\square$

### A.4. Section 4.4

We have that $\bar\rho(\bar g(x), w) = \sum_{\ell=1}^{L} \rho(g^{(\ell)}(x^{(\ell)}), w^{(\ell)})$ decomposes as

$$\bar\rho(\bar g(x), w) = \sum_{\ell=1}^{L} \rho(g^{(\ell)}(x^{(\ell)}), w^{(\ell)})$$

$$= \sum_{\ell=1}^{L}\sum_{i=1}^{k}\sum_{j_1,j_2,j_3\in I_{3,d_\ell}} \operatorname{Re}\left(A_{i,j_1}^{(\ell)} A_{i,j_2}^{(\ell)} (A_{i,j_3}^{(\ell)})^2\right) \frac{x_{j_1}^{(\ell)} x_{j_2}^{(\ell)} (w_{j_3}^{(\ell)})^2}{k}$$

$$= \sum_{\ell=1}^{L}\sum_{i=1}^{k}\sum_{j_1,j_2,j_3\in I'_{3,d}} \operatorname{Re}\left(A_{i,j_1}^{(\ell)} A_{i,j_2}^{(\ell)} (A_{i,j_3}^{(\ell)})^2\right) x_{j_1}^{(\ell)} x_{j_2}^{(\ell)} (w_{j_3}^{(\ell)})^2/k$$

$$+ 2\sum_{\ell=1}^{L}\sum_{i=1}^{k}\sum_{j_1,j_2\in I'_{2,d}} \operatorname{Re}\left(A_{i,j_1}^{(\ell)} (A_{i,j_2}^{(\ell)})^3\right) x_{j_1}^{(\ell)} x_{j_2}^{(\ell)} (w_{j_2}^{(\ell)})^2/k$$

$$+ \sum_{\ell=1}^{L}\sum_{i=1}^{k}\sum_{j_1,j_2\in I'_{2,d}} \operatorname{Re}\left((A_{i,j_1}^{(\ell)})^2 (A_{i,j_2}^{(\ell)})^2\right) (x_{j_1}^{(\ell)})^2 (w_{j_2}^{(\ell)})^2/k$$

$$+ \sum_{\ell=1}^{L}\sum_{i=1}^{k}\sum_{j=1}^{d} w_j^2 x_j^2$$

$$= \sum_{\ell=1}^{L}\sum_{i=1}^{k} H_{1,\ell,i} + \sum_{\ell=1}^{L}\sum_{i=1}^{k} H_{2,\ell,i} + \sum_{\ell=1}^{L}\sum_{i=1}^{k} H_{3,\ell,i} + \|x\|_w^2$$

$$= H_1 + H_2 + H_3 + \|x\|_w^2.$$

For ease of exposition, we only deal with the decomposition of $H_3$, as the discussion for $H_1$ and $H_2$ is similar. We first bound some quantities related to the Orliz norms of the Rademacher chaoses $H_{3,\ell,i}$.

**Lemma A.4.** *Let* $b = (\|H_{3,\ell,i}\|_{\psi_\alpha} : \ell = 1,\dots,L;\; i = 1,\dots,k)$. *We have that, for some constant* $\bar c$,

$$\|b\|_2 \le \bar c\,\|w\|_4^2\,\|x\|_2^2/(\sqrt{k}L)$$

*and that*

$$\|b\|_\infty \le \bar c\,\|w\|_4^2\,\|x\|_2^2/(kL^{3/2}).$$

*Proof.* We have that

$$\|b\|_2^2 = \sum_{\ell=1}^{L}\sum_{i=1}^{k} \|H_{3,\ell,j}\|_{\psi_\alpha}^2 \le 24e^2(e+1)^{2/\alpha} \sum_{\ell=1}^{L}\sum_{i=1}^{k} W(H_{3,\ell,j})^2$$

$$\le 24e^2(e+1)^{2/\alpha}\zeta \sum_{\ell=1}^{L}\sum_{i=1}^{k} \|x_\ell\|_2^4\,\|w_\ell\|_4^4\,k^{-2} \le \bar c\frac{\|w\|_4^4\,\|x\|_2^4}{k^2 L^3}kL$$

Where we used Lemma 3.3 with the fact that $x$ and $w$ are near-uniform.

Similarly,

$$\|b\|_\infty^2 = \max_{\ell=1,\ldots,L} \max_{i=1,\ldots,k} \|H_{3,\ell,j}\|_{\psi_\alpha}^2 \le \overline{c} \max_{\ell=1,\ldots,L} \max_{i=1,\ldots,k} \|x_\ell\|_2^4 \|w_\ell\|_4^4 k^{-2} \le \overline{c} \frac{\|w\|_4^4 \|x\|_2^4}{k^2 L^3}.$$

$\square$

We then have the following result.

**Lemma A.5.** *We have* $\Pr\left[|H_3| \ge \|w\|_4^2 \|x\|_2^2 \bar{F}(t)\right] \le 2e^{-t} \ \forall t > 0$, *with* $\bar{F}(t) = \frac{c_1}{\sqrt{kL}} \sqrt{t} + \frac{c_2}{kL^{3/2}} t^2$ *and* $c_1$ *and* $c_2$ *appropriate constants.*

*Proof.* Recall that $H_1 = \sum_{\ell=1}^L \sum_{i=1}^k H_{3,\ell,i}$, with each summand a mean-zero Rademacher chaos of order at most 4. We then have (Kuchibhotla & Chakrabortty, 2022) for suitable constants $\overline{c_1}$ and $\overline{c_2}$ that

$$\Pr\left[|H_3| \ge \overline{c_1} \|b\|_2 \sqrt{t} + \overline{c_2} \|b\|_\infty t^2\right] \le 2e^{-t}.$$

Plugging in the result of the previous lemma, we have:

$$2e^{-t} \ge \Pr\left[|H_3| \ge \overline{c_1} \|b\|_2 \sqrt{t} + \overline{c_2} \|b\|_\infty t^2\right] \ge \Pr\left[|H_3| \ge \overline{c_1} \frac{\overline{c} \|w\|_4^2 \|x\|_2^2}{\sqrt{kL}} \sqrt{t} + \overline{c_2} \frac{\overline{c} \|w\|_4^2 \|x\|_2^2}{kL^{3/2}} t^2\right]$$

Choosing $c_1 = \overline{c_1}\overline{c}$ and $c_2 = \overline{c_2}\overline{c}$ gets the claim. $\square$

Similar results hold for $H_1$ and $H_2$. We then have the following.

**Lemma A.6.** *We have that* $\Pr\left[|H| \ge 4 \|w\|_4^2 \|x\|_2^2 \bar{F}(t)\right] \le 8e^{-t}$.

*Proof.* We use union bound over $H_1$, $H_2$, and $H_3$, similarly to what is done in Lemma 4.2. $\square$

We can then prove the lemma stated in the main paper.

**Lemma 4.5.** *For any given* $x, w \in \mathbb{R}^d$, *we have*

$$\Pr\left[|\bar{\rho}(\bar{g}(x), w) - \|x\|_w^2| > t\right] \le \max \begin{cases} 8\exp\left(-kc_1 \frac{L^2}{(\|w\|_4^2\|x\|_2^2)^2} t^2\right) \\ 8\exp\left(-\sqrt{k}c_2 \frac{L^{3/4}}{(\|w\|_4^2\|x\|_2^2)^{1/2}} \sqrt{t}\right) \end{cases}$$

*with* $c_1$ *and* $c_2$ *appropriate constants (different from the ones in Lemma A.5).*

*Proof.* We have

$$P = \Pr\left[|\bar{\rho}(\bar{g}(x), w) - \|x\|_w^2| > t\right] = \Pr\left[|H| > t\right].$$

From Lemma A.6, we have that

$$\Pr\left[|H| > 4c_1 \frac{\|w\|_4^2 \|x\|_2^2}{L\sqrt{k}} \sqrt{v} + 4c_2 \frac{\|w\|_4^2 \|x\|_2^2}{L^{3/2}k} v^2\right] \le 8e^{-v}.$$

Then, for $v$ such that the term with $\sqrt{v}$ dominates, we have that $P \le \Pr\left[|H| > 8c_1 \frac{\|w\|_4^2 \|x\|_2^2}{L\sqrt{k}} \sqrt{v}\right] \le 8e^{-v}$.

Otherwise, we have $P \le \Pr\left[|H| > 8c_2 \frac{\|w\|_4^2 \|x\|_2^2}{L^{3/2}k} v^2\right] \le 8e^{-v}$.

Then, setting $v$ accordingly yields the claim. $\square$

**Lemma 4.6.** Let $\varepsilon, \delta > 0$. Let $c$ be a suitable universal constant. Let $L$ and $k$ be positive integers such that $k \geq c \max \left\{ \frac{\Delta^2 \ln(8/\delta)}{L^2 \varepsilon^2}, \frac{\Delta \ln(8/\delta)^2}{L^{3/2} \varepsilon} \right\}$. Then there exists a *linear* function $g(x) : \mathbb{R}^d \to \mathbb{C}^{k \cdot L}$ and an estimator $\rho(g(x), w) : \mathbb{C}^{k \cdot L} \times \mathbb{R}^d \to \mathbb{R}$ such that for any given $x, w \in \mathbb{R}^d$, with probability at least $1 - \delta$,

$$\left| \rho(g(x), w) - \|x\|_w^2 \right| < \varepsilon \|x\|_2^2 \|w\|_4^2 / \Delta.$$

In particular, if $\|x\|_2^2 \|w\|_4^2 / \|x\|_w^2 \leq \Delta$, we get, with probability at least $1 - \delta$,

$$(1 - \varepsilon) \|x\|_w^2 < \rho(g(x), w) < (1 + \varepsilon) \|x\|_w^2.$$

*Proof.* Let $t = \varepsilon \|w\|_4^2 \|x\|_2^2 / \Delta$.

Suppose we are in the Gaussian tail case of the bound. Then we have, taking $k = c_1^{-1} \Delta^2 \varepsilon^{-2} \ln(8/\delta) L^{-2}$,

$$\Pr \left[ |\rho(g(x), w) - \|x\|_w^2 | > t \right] \leq 8 \exp \left( -k c_1 \frac{t^2 \cdot L^2}{(\|w\|_4^2 \|x\|_2^2)^2} \right) = \delta.$$

Let now assume we are in the long tail part of the bound. Then, taking $k = c_2^{-2} \Delta \varepsilon^{-1} \ln(8/\delta)^2 L^{-3/2}$,

$$\Pr \left[ |\rho(g(x), w) - \|x\|_w^2 | > t \right] \leq 8 \exp \left( -c_2 \sqrt{k} \frac{\sqrt{t} L^{3/4}}{(\|w\|_4^2 \|x\|_2^2)^{1/2}} \right) = \delta.$$

As the failure probability decreases by $k$, by taking $k \geq c \max \left\{ \frac{\Delta^2 \ln(8/\delta)}{L^2 \varepsilon^2}, \frac{\Delta \ln(8/\delta)^2}{L^{3/2} \varepsilon} \right\}$, with an appropriate constant $c$, completes the proof. $\square$

# B. Experiments

We perform here a proof-of-concept experimental evaluation of our techniques. We generate a random vector $x \in \mathbb{R}^d$ by selecting uniformly at random each entry in $[-1, 1]$ and then normalizing the resulting vector such that $\|x\|_2 = 1$. The random weight vector $w \in \mathbb{R}^d$ is also created by selecting uniformly at random each entry in $[0.1, 1.2]$. We set $d = 2 \cdot 10^4$. Finally, we generate 1000 random matrices $A$ and plot the empirical distribution of the estimator $\rho(g(x), w)$.

From the experiments, we see a gap in the quality of the estimate between the linear (Figure 2) and nonlinear (Figure 1) method by Kaban (2015), as argued in Section 4.3. Moreover, the experimental results in Figure 5 showcase the applicability of the method on sparse vectors, as discussed in Section 4.3, as well as the poor quality of the estimate for near-uniform vectors for the method described in Theorem 1.2, suggesting that the probabilistic analysis of the method is tight. Finally, the experiments in Figures 4 and 3 highlight the significant reduction of the variance of the estimator when using the sparse block matrices. Interestingly, the sparse map with the decomposition into $L$ sub-vectors almost closes the gap with the non-linear method, as argued in Section 4.5.

Our code is publicly available at https://github.com/aidaLabDEI/complex-dimensionality-reduction.

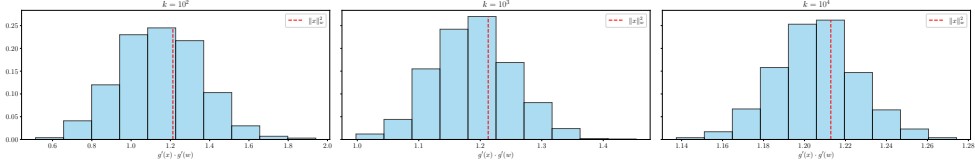

*Figure 1.* Empirical distribution of the *nonlinear* estimator $g'(x) \cdot g'(w)$ based on JL described in Equation 1, over $N = 10^3$ random matrices $A$. The vectors $x$ and $w$ are fixed throughout the experiment, and drawn from a uniform distribution on $[-1, 1]^d$ and on $[1.0, 1.2]^d$, respectively. The dimensionality of the vectors is $d = 2 \cdot 10^4$. We report the results for $k \in \{10^2, 10^3, 10^4\}$. We remark that, due to its nonlinearity, this method does not allow to compute pairwise distances.

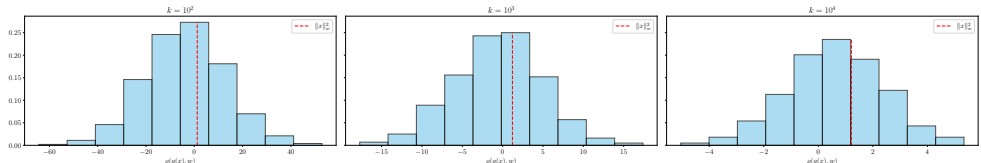

*Figure 2.* Empirical distribution of the *linear* estimator $\rho(x, w)$ described in Theorem 1.1, over $N = 10^3$ complex random matrices $A$. The vectors $x$ and $w$ are fixed throughout the experiment, and drawn from a uniform distribution on $[-1, 1]^d$ and on $[1.0, 1.2]^d$, respectively. The dimensionality of the vectors is $d = 2 \cdot 10^4$. We report the results for $k \in \{10^2, 10^3, 10^4\}$. We note that the choice of $x$ and $w$ is the worst-case scenario discussed in Section 4.3, providing experimental evidence for the tightness of the analysis.

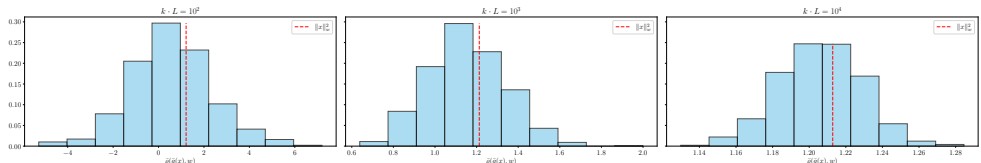

*Figure 3.* Empirical distribution of the *linear* estimator $\bar{\rho}(x, w)$ described in Theorem 4.6, over $N = 10^3$ complex random matrices $A$. The vectors $x$ and $w$ are fixed throughout the experiment, and drawn from a uniform distribution on $[-1, 1]^d$ and on $[1.0, 1.2]^d$, respectively. The dimensionality of the vectors is $d = 2 \cdot 10^4$. We report the results for $L \in \{10^2, 10^3, 10^4\}$ and $k = 1$, which is the optimal parameter choice given the almost-uniformity of the vectors $x$ and $w$. The results showcase the reduction of the estimator variance when using the vector partitioning technique, thereby closing the gap with the nonlinear method.

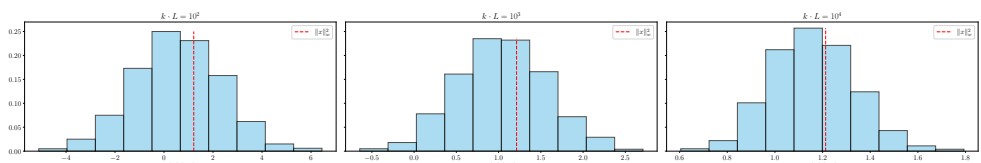

*Figure 4.* Empirical distribution of the *linear* estimator $\bar{\rho}(x, w)$ described in Theorem 4.6, over $N = 10^3$ complex random matrices $A$. The vectors $x$ and $w$ are fixed throughout the experiment, and drawn from a uniform distribution on $[-1, 1]^d$ and on $[1.0, 1.2]^d$, respectively. The dimensionality of the vectors is $d = 2 \cdot 10^4$. We report the results for $L = 100$ and $k \cdot L \in \{10^2, 10^3, 10^4\}$, which ensures the almost-uniformity condition for a wider range of vectors $x$ and $w$.

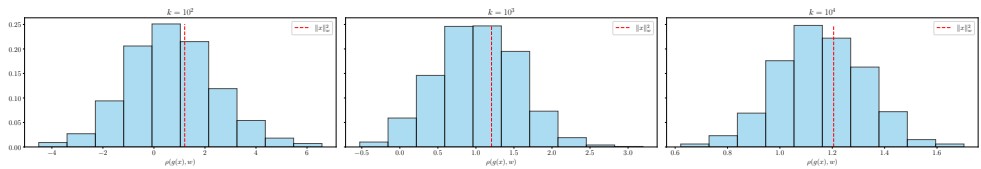

*Figure 5.* Empirical distribution of the *linear* estimator $\rho(x, w)$ described in Theorem 1.1, over $N = 10^3$ complex random matrices $A$. The vector $x$ is sparse, with $m = 10^2$ entries drawn from a uniform distribution on $[-1, 1]^d$ and the rest set to 0. Similarly, $w$ is sparse, with the same entries drawn from a uniform distribution on $[1.0, 1.2]^d$. The dimensionality is $d = 2 \cdot 10^4$. We report the results for $k \in \{10^2, 10^3, 10^4\}$. We note that the concentration depends on $m$, rather than on $d$, as argued in Section 4.3.

