# OpenReview forum: "Dimensionality Reduction on Complex Vector Spaces for Euclidean Distance with Dynamic Weights"
_ICML.cc/2025/Conference — ICML 2025 poster_

### Official Review · Reviewer_xjUY · 2025-03-10

**Overall Recommendation:** 3

**Summary:**

This paper presents an embedding of a d-dimensional vector x into k dimension such that, for any d-dimensional weight vector w $\sum_{i=1}^d w_i^2 x_i^2$ is preserved. Specifically, the authors give an additive error in terms of $\varepsilon \|x\|_2^2 \cdot \|w\|_4^2$. The classic Johnson-Lindenstrauss guarantee for unweighted vectors, that is, $w_i=1$ for all $i$, has an additive error of $\varepsilon \|x\|_2^2$, which leads to a multiplicative $(1\pm \varepsilon)$ approximation.

Traditionally, the entries matrices with Johnson-Lindenstrauss moments are subsampled Rademacher random variables. The authors extend this construction by choosing the embedding entries to be complex numbers, where the sign of the complex part is Rademacher distributed. This sketch is applied to input vectors $x$, and is then given to a decoding algorithm. The construction is novel and the use of complex numbers is clever. After this setup, the analysis is not too difficult, but requires controlling stochastic processes in ways that are not standard.

**Claims And Evidence:**

-

**Essential References Not Discussed:**

-

**Experimental Designs Or Analyses:**

There are no experiments. Comparing the performance between the author's embedding and the result by Kaban would have been nice, if only to also see whether the theoretical bounds could be improved or whether there is a gap between linear and non-linear methods. It would have also been nice to see if the performance is sensitive to the sparsity of the JL-matrix used (e.g. if the peformance is the same if a constant fraction of entries are set to 0).

**Methods And Evaluation Criteria:**

-

**Other Comments Or Suggestions:**

-

**Other Strengths And Weaknesses:**

I find the ideas quite clever and the technique engaging.

Nevertheless, it would have been nice if the result had achieved an optimal error, or if the analysis had been tight, or if the authors had presented a framework for generalizing general JL-embedding matrices, rather than only the complex analogue of dense Rademacher sketches. I understand that it might be difficult to improve the paper in the way that I am suggesting.

Alternatively, some experimental focus would have been welcome for a conference like ICML.

In the end, I feel this paper barely gets a weak accept. For a higher rating, it would need to address one of the aforementioned points in a satisfactory way.

**Questions For Authors:**

-

**Relation To Broader Scientific Literature:**

I believe that this result is a bit niche. Why a linear method is important for applications still eludes me. In addition, instead of developing a general framework for JL-transforms into a complex space with a range of sparsity/target dimension tradeoffs, the authors give one construction for which it is not clear that the presented analysis is tight.

I think a compelling use case for which the authors offer a real improvement over the state of the art, a tight analysis of their method, or alternatively, but unlikely, a separation between linear and non-linear methods, or a general framework for sparse JL transforms into complex spaces would have increased my score substantially.

**Theoretical Claims:**

I have previously reviewed the paper. While there are some changes, the correctness proofs are the same and I still believe the results to be correct.

---

> ### Author Rebuttal · Authors · 2025-03-31
>
> We thank Reviewer 3 for the useful feedback. We provide below answers to reviewer’s comments.
>
> ---
>
> **There are no experiments. Comparing the performance with the result by Kaban would have been nice, if only to also see whether the theoretical bounds could be improved.**
>
> Thank you for the suggestion. Based also on your comment, we added some proof of concept experiments where we show the empirical distribution of the estimates $\rho$ of the weighted norm.
>
> Link: www.dropbox.com/scl/fo/lvallzefg646aayc826zv/ALbK6BOw0u5PANsePzCDGsM?rlkey=tns4b16zt86ce5tmjpdxdrbnh
>
> In particular, the experiments highlight (Fig 3) the significant reduction of the variance of the estimator when using the technique described in Sect 4.4, which uses sparse block matrices. Notably, this technique introduces sparseness (via decomposing the vector into $L$ sub-vectors) to reduce the variance, and the trade-off lies in ensuring that the number of sub-vectors $L$ is small enough that the norm of each one is roughly $1/L$ of the total norm. We will highlight this tradeoff in the paper.
> Moreover, the experimental results showcase the applicability of the sparse method (Fig 5), as discussed in Sect 4.3, as well as the poor quality of the estimate for near-uniform vectors for the method described in Thm 1.1 (Fig 2), suggesting that the probabilistic analysis of the method is tight.
> Finally, from the experiments we indeed see a gap in the quality of the estimate between the linear and nonlinear (Kaban, Fig 1) method, as argued in Sect 4.3. Interestingly, the sparse map with the decomposition into $L$ sub-vectors almost closes the gap with the non-linear method.
>
> **Why a linear method is important for applications still eludes me.**
>
> Thank you for pointing out this lack of clarity. The main feature of linearity is the ability to compute distances between vectors by simply computing distances between reduced vectors, since $\lVert x - y \lVert \simeq \lVert g(x - y) \lVert = \lVert g(x) - g(y) \lVert$. Based on your comment, we will include the following paragraph to make this clearer.
>
> > Indeed, the prime application of the linearity of JL maps is reducing the time complexity of pairwise distance computations [1], which has applications in clustering [2] and nearest neighbor search [3, 7], e.g. for recommender systems and classification. Other examples include compressed sensing [4], which relies on sparse linear measurements to recover an unknown signal. Finally, the linearity of JL has been exploited to speed up least square regression [5] and low-rank matrix approximation [6, 1].
>
> (See answer to Reviewer 1 for the new references [1-7]).
>
> **A compelling use case that offers a real improvement over SOTA.**
>
> We first observe that our approach represents the first linear dimensionality reduction supporting weighted case. Moreover, we partially address the sparse JL transformation by showing in Section 4.4 that a sparse block structure allows to reduce variance in case of almost uniform vectors.
>
> **Provide [...] a tight analysis of their method. I understand that it might be difficult to improve the paper...**
>
> Thank you for your suggestion. We would indeed like to find a lower bound, but we were so far unable to obtain one. We conjecture there is some form of information-theoretic lower bound, as arbitrary approximations of the weighted norm would allow to reconstruct the vector at hand.
>
> We can however provide a strong hint that our probabilistic analysis is indeed tight. Note that this is not a formal proof.
> We have, by [1, Example 3.3] and  [2, Thm 1.1], that
> $\lVert \sum_{i=1}^k S_i \rVert_q = \Theta\Big( \big( \sum_{i=1}^k \lVert S_i \lVert_q^q \big)^{1/q} + \big( \sum_{i=1}^k q \lVert S_i \lVert_2^2 \big)^{1/2}  \Big) \geq  C \sqrt {kq} \cdot W(S_1)$ for some universal constant $C$.
>
> We take $x = (1, \dots, 1) \in \mathbb{R}^d$ and $w = (1, \dots, 1) \in \mathbb{R}^d$. Note that $\lVert x \lVert^2_2 \lVert w \lVert^2_4 = d^{3/2}$. Then, in the proof of Lemma A.1, we have that $W(H_{1,1,i}) = |I'_{d,3}| \geq d^{3/2} / 30$ if $d > 3$. This then implies that $\lVert H_1 \lVert_q \geq C \sqrt {kq} \cdot  d^{3/2}$ for any $q > 2$, for some universal constant $C$.
>
> If we suppose that we can obtain an upper bound on the tail probabilities of the type $\mathbb{P}[|H_1| > t] \leq \exp( - c (t / f(d) )^{\alpha} )$, with $\alpha > 0$, we obtain that $\lVert H_1 \lVert_q \leq C' f(d)$, for some universal constant $C'$ that depends on $q$ but not $d$. Therefore, if we take $f(d)\in o(d^{3/2})$, for large enough $d$ it would violate the condition found above, and we have a contradiction.
> In conclusion, the dependence of our bounds on $\lVert x \lVert^2_2 \lVert w \lVert^2_4 = d^{3/2}$ seems to be optimal.
>
> - [1] Latała. "Estimation of moments of sums of independent real random variables" The Annals of Probability.
> - [2] Hitczenko et al. "Moment inequalities for sums of certain independent symmetric random variables." Studia Math.

---

> > ### Comment · Reviewer_xjUY · 2025-04-07
> >
> > I think it may have not been clear what I meant by "important for applications" or an improvement over the SOTA. I understand how linear transforms are beneficial in various models. What I wanted to know is whether there exists a problem which uses estimates for the weighted Euclidean norm as a primitive that obtains an improvement via this work. The references that you provide showcase that sketching is important, but none of these applications require the weighted norm. Indeed, for problems like regression or low rank approximation, weighted norms make little sense, at least to me.

---

> > > ### Author Response · Authors · 2025-04-08
> > >
> > > We had misunderstood the "improvement", as our method is the first of its kind. We now see you meant in applications.
> > >
> > > We sketched in the paper (e.g. lines 28-38) applications of the weighted norms in low rank approx. (e.g. Woodruff et al. 2024), recommender systems (e.g. Wang et al. 2015; Gu et al. 2016) and k-NN classifiers (e.g. [1], and Moreo et al. 2020, Bhattacharya 2017).
> > >
> > > We indeed agree with you that an important application of our technique would make it more impactful, thanks for the suggestion.
> > > We will discuss the following applications, where our technique can yield some improvements:
> > >
> > > **Weighted least squares**
> > >
> > > We consider the weighted least squares problem [5, Chapter 6], which provides robust estimators in the presence of uneven reliability in the measurement [6], where given a data matrix $X \in \mathbb{R}^{n \times d}$, an observation vector $y \in \mathbb{R}^n$, and a weight vector $w \in \mathbb{R}^n$ (with $w_i \ge 0$ for all $i$), the goal is to find $\theta \in \mathbb{R}^d$ that minimizes the weighted norm $\lVert X \theta - y\rVert_w^2 = \sum_{i=1}^n w_i^2 (X_i^\top \theta - y_i)^2$. Solving this problem (e.g. by QR decomposition) takes $O(n d^2)$ time.
> > > One possible approach when $n \gg d$ is to form the weighted data by defining $X' = WX$ and $y' = Wy$, where $W$ is the diagonal matrix with entries $w_i$, and then to apply a standard Johnson–Lindenstrauss (JL) transform to reduce the problem dimension [7]. As noted in the introduction, this though requires the weights to be known before the dimensionality reductions, and does not allow to solve the problem with multiple weights.
> > >
> > > In contrast, we sketch the use our weighted JL-like construction as follows. We consider the linear map $g(x)$ that reduces any vector $x$ to a lower-dimensional one and and the norm estimator $\rho(g(x), w)$ given by Theorem 1.1. Then, the reduced data matrix $g(X) \in \mathbb{C}^{k\times d}$ and observation vector $g(y) \in \mathbb{C}^k$ are computed in $O(kdn)$ time, before the weight vector is revealed.
> > > Then, the weighted least squares objective $\lVert X \theta - y \rVert_w^2$ is approximated by the weighted norm oracle $F(\theta) = \rho(g(X)\theta-g(y), w)$, which by linearity of $g$ equals $\rho(g(X\theta-y), w)$, and an iterative optimization method (e.g. Nelder-Mead) is used to update $\theta$ based on low-dimensional evaluations of $F(\theta)$, which take $O(kd)$ time each. This allows to solve least squares for several weight vectors, possibly amortizing the cost of computing the reduced representations.
> > >
> > > **Applications in machine learning**
> > > - In [2], the authors propose a time-efficient alternative to the dynamic time warping metric for time series based on the Mahalanobis distance with diagonal covariance matrix, which is the weighted euclidean distance. When the distance is used for classification, a weight vectors is learned on a per-class basis, resulting in a set of different weights vectors. Since the complexity of the distance computation scales linearly with the time series' length, this can become computationally challenging, especially in a steaming setting. Our dimensionality reduction technique allows to compress the time series into reduced vectors, and subsequently compute the weighted distance between such time series, with any given weight vector.
> > > - Relatedly to the previous application, in [3] and [4] the authors propose kernel methods where the weighted distance (phrased as a Mahalanobis distance with diagonal covariance matrix) is used as a substitute for the plain Euclidean distance in RBF-like kernels. Our techniques would then allow to apply such kernels on the reduced vectors rather than on the original data, with any weight vectors (e.g. for two different tasks).
> > >
> > > ----
> > >
> > > We also note that in ICML 2019,  Lei et al. started the study of data structures supporting nearest neighbor searches on weighed Euclidean metrics, so the field of weighted norms seems to be developing.
> > >
> > > Finally, we hope that our new experimental results and the discussion on the tightness of our analyis addressed your other concerns (if so, we'd be grateful if you updated your score!).
> > >
> > > ----
> > > .
> > >
> > > - [1] R. Paredes and E. Vidal, Learning Weighted Metrics to Minimize Nearest-Neighbor Classification Error. In IEEE TPAMI, 2006
> > > - [2] Prekopcsák, Z., Lemire, D. Time series classification by class-specific Mahalanobis distance measures. Adv Data Anal Classif 6,  2012.
> > > - [3] Abe, S. (2005). Training of Support Vector Machines with Mahalanobis Kernels. In: Artificial Neural Networks: Formal Models and Their Applications
> > > - [4] Kamada, Y., Abe, S. (2006). Support Vector Regression Using Mahalanobis Kernels. In: Artificial Neural Networks in Pattern Recognition.
> > > - [5] Golub, G. H., & Van Loan, C. F. (2013). Matrix Computations.
> > > - [6] Fox, J. (2015). Applied Regression Analysis and Generalized Linear Models.
> > > - [7] Sarlos, T. Improved approximation algorithms for large matrices via random projections. FOCS 2006.

---

### Official Review · Reviewer_pXLS · 2025-03-12

**Overall Recommendation:** 3

**Summary:**

This paper explores dimensionality reduction on complex vectors for Euclidean distances. The authors decompose the complex dimensionality reduction into several Rademacher chaos random variables, where novel concentration inequalities for sums of independent Rademacher chaoses are derived.

**Claims And Evidence:**

The claims seem to be reasonable.

**Essential References Not Discussed:**

The references are sufficient.

**Experimental Designs Or Analyses:**

No experiments are provided.

**Methods And Evaluation Criteria:**

There are no experiments.

**Other Comments Or Suggestions:**

Few experiments should be provided.

**Other Strengths And Weaknesses:**

This paper provides a theoretical framework for finding linear functions in the case of weighted L2 norm. For weaknesses, please refer to the problem below.

**Questions For Authors:**

1)The title contains dynamic weights. It seems that the authors do not mention them in detail in this paper.

2)The weighted L2  norm is explored. It is meaningful to study the general norm.

3) Introducing complex vector spaces is interesting. What is the effect of the complex vector spaces when devising specific algorithms?

4) It would be much better to use toy examples to show the probability bound. Maybe a specific example for the privacy-preserving similarity search should be given.

5) What is the near-uniform vector?

**Relation To Broader Scientific Literature:**

This paper provides a theoretical framework for finding linear functions in the case of weighted L2 norm.

**Theoretical Claims:**

I checked the proofs of some theorems, and they sound correct.

---

> ### Author Rebuttal · Authors · 2025-03-31
>
> We thank Reviewer 2 for the useful feedback. We provide below detailed answers to comments.
>
> ----
>
> **Few experiments should be provided.**
>
> Thank you for the suggestion, we added some proof of concept experiments where we show the empirical distribution of the estimates $\rho$ of the weighted norm. See comment of Reviewer 3 for details.
>
> Link: www.dropbox.com/scl/fo/lvallzefg646aayc826zv/ALbK6BOw0u5PANsePzCDGsM?rlkey=tns4b16zt86ce5tmjpdxdrbnh
>
> **The title contains dynamic weights but are not mentioned in detail**
>
> Based on your feedback, we will expand the introduction (e.g. lines 28-38, second column) with the following sentences and references to highlight the practical relevance of weighted norm queries where the weights can change dynamically from one query to another.
>
> > However, weights might change at each query, or they might not known at preproccessing time, a setting that we hereinafter refer to as “dynamic weights”. Examples include nearest neighbor methods [1] for classification or recommender systems [2], where the relative importance of features might depend on the type of query at hand.  Weights might be estimated on highly dynamic datasets which require a frequent update of the parameters (e.g., (Banihashem et al., 2024)).
>
>  - [1] Indyk, Piotr, and Rajeev Motwani. "Approximate nearest neighbors: towards removing the curse of dimensionality." STOC 1998.
>  - [2] Baumgärtner, Tim, et al. "Incorporating Relevance Feedback for Information-Seeking Retrieval using Few-Shot Document Re-Ranking." EMNLP 2022.
>
> **What is the effect of the complex vector spaces?**
>
> The intuition on why a complex space is useful is that it allows to remove the bias in the estimator for $\| x \|^2_w$ when we have terms in it with three or more multiplicands.
> More formally, the key observation, which can be found in Theorem 4.1, is that the contribution of $x_{j_1} x_{j_2} w^2_{j_3}$ in the estimator is multiplied by the random variable $A_{i, j_1} A_{i, j_2} A^2_{i, j_3}$. Since each entry of $A$ is iid in $\{\pm 1, \pm i\}$, we have that the r.v. is a uniform on either $\{\pm 1, \pm i\}$ or on $\{\pm 1\}$ when any indices of $j_1, j_2, j_3$ are distinct (and thus it is $0$ in expectation), and the constant $1$ when $j_1, j_2, j_3$ are all the same, since $\pm 1^4 = \pm i^4 = 1$. Then, in expectation only the $x_{j} x_{j} w^2_{j}$'s contribute to the estimator.
>
> **It is meaningful to study the general norm.**
>
> This is a very intriguing remark. We indeed agree that exploring norms beyond the $L_2$ would be of great interest, albeit not trivial: we believe that the approach can be generalized to $L_p$ by using roots of unit of higher order. Based on your comment, we will introduce the following remark to guide future work in this direction.
>
> >We conjecture that the map to the complex plane we provide here admits a generalization to the estimation of arbitrary (weighted) $L_p$ norms. Indeed, let $h = \sum_{j_1, \dots j_p}\prod_{k=1}^p A_{j_k}x_{j_k}$. If $A$ is a uniform on the $p$-th roots of $1$, we have that $\mathbb{E}[\prod_{k=1}^p A_{j_k}] = 1$ if all the $j_k$'s are the same, and $0$ otherwise. Therefore, $\mathbb{E}[ h ] = \| x \|_p^p$. However, the variance of the estimator grows with $p$, and novel approaches should be developed in order to reduce this growth.
>
> **Use toy examples to show the probability bound and the privacy-preserving similarity search**
>
> We added some proof-of-concept experiments on some toy data to show the distribution of the estimator. See the reply to the first comment and Reviewer 3.
>
> An example of privacy-preserving similarity search is the following: release a dataset, so that a user can detect if there exists a point $x$ within distance $r$ from a given query $q$, but no information on the exact values of $x$ are released. JL provides a solution under differential privacy (Blocki et al, FOCS 12). We believe that an interesting extension would be to release a dataset so that a user can detect if there are points within the desired weighted norm without revealing the exact point coordinates. Indeed, the variance of the estimates would avoid the reconstruction of the original vectors.
> We plan to add the following sentence (at line 434, second column):
>
> >Finally, we believe that our constructions can be of interest for privacy preserving similarity search as it might allow the release of datasets that allow users to detect if there are near points within the desired weighted norms without releasing the details of the vector, similar to what done in  (Blocki et al, FOCS 12) for JL.
>
> **What is the near-uniform vector?**
> A uniform vector is a vector where all entries are the same. A near uniform vector is a relaxation of this condition, where splitting the vector into contiguous parts results in sub-vectors with roughly the same norm. We give a formal definition in the beginning of Section 4.4, and we will insert the informal description above in the paper to clarify it.

---

### Official Review · Reviewer_oog7 · 2025-03-14

**Overall Recommendation:** 3

**Summary:**

The main result of the paper is the following (Theorem 1.1):

Let $\epsilon, \delta  \in (0,1)$ and $\Delta \ge 0$ be given parameters. There is a function $g : \mathbb{R}^d \rightarrow \mathbb{R}^{O(\Delta^2\log(1/\delta)/\epsilon^2)}$ and an estimator $\rho(g(x), w)$ such that for any vectors $x,w \in \mathbb{R}^d$, with prob. $1-\delta$
$$ \left | \left(\sum_{i=1}^d x_i^2 w_i^2 \right) - \rho(g(x),w) \right| \le \epsilon \cdot \|x\|_2^2 \cdot  \|w\|_4^2/\Delta. $$

The authors seem to have missed the fact that the something similar to the main theorem easily follows from a one line calculation from the standard Johnson-Lindenstrauss lemma. Let me give the full proof below:

Let $\Pi: \mathbb{R}^d\rightarrow \mathbb{R}^{O(\log(1/\delta)/\epsilon^2)}$ be a JL map. It is well known that for any vectors $a,b$, with prob. $1-\delta$, we have

$$ | \langle \Pi a, \Pi b \rangle - \langle a, b\rangle  | \le \epsilon \|a \|_2 \|b\|_2.$$

(This is a standard exercise, e.g. see hw 1 here:  https://rajeshjayaram.com/homeworks/hw2.pdf).

Now apply this to the vectors $x' = [x_1^2, \ldots, x_d^2]$ and $w' = [w_1^2, \ldots, w_d^2]$. We have $\sum_{i=1}^d x_i^2 w_i^2  = \langle x', w' \rangle$ so

$$ \left | \left(\sum_{i=1}^d x_i^2 w_i^2 \right) - \langle \Pi x', \Pi w' \rangle \right| \le \epsilon \cdot \|x'\|_2  \|w'\|_2 \le \epsilon \|x\|_2^2 \| w\|_2^2. $$
Then we can scale $\epsilon$ by any $\Delta \ge 0$ to get an error guarantee of $\epsilon \|x\|_2^2  \|w\|_2^2/\Delta$ by projecting $x$ onto $O(\log(1/\delta) \Delta^2 /\epsilon^2)$ dimensions. This version is slightly weaker than the main theorem since we replace the $4$ norm with $2$, but better in the sense that we don't have to take the max over two dimensions that is done in Theorem 1.1.

However, I can't imagine a situation where replacing the $4$ norm by $2$ would play a big role. This argument also has the benefit that it fits into a few lines with known tools rather than many pages with complicated analysis. Furthermore this construction seems easier to implement and is arguably much more practical than the "radamacher chaos" based construction of the paper.

Update: After reading the rebuttal, I have updated the score.

**Claims And Evidence:**

Yes, the proofs seem correct although quite complicated.

**Essential References Not Discussed:**

None.

**Experimental Designs Or Analyses:**

No experiments.

**Methods And Evaluation Criteria:**

No empirical results.

**Other Comments Or Suggestions:**

None

**Other Strengths And Weaknesses:**

Are there any realistic scenarios where the construction of the paper is to be preferred over the simpler construction outlined above?

**Questions For Authors:**

None.

**Relation To Broader Scientific Literature:**

None.

**Theoretical Claims:**

The proofs seem correct but the calculations are long so I was not able to follow them directly. However, the simpler argument I gave above is immediate.

---

> ### Author Rebuttal · Authors · 2025-03-31
>
> We are very confused by this review, and we believe that it might be due to a misunderstanding.
> The summary that the reviewer provides is misleading and incomplete, as in his re-formulation of our main result, Theorem 1.1, they crucially omit that the function $g$ is linear. Because of this, the reviewer then describes the construction of a nonlinear function, which is a fundamentally different goal to the one we pose in this paper (e.g., because it does not allow to compute distances between vectors). In fact, we indeed explicitly address the proposed nonlinear construction in the Introduction section (lines 75-92 first column, e.g. Equation 1) of our submission.
>
>
> We remark that linearity is a desiderable property also in practical scenarios. In order to make this even more clear to the reader, we will add the following sentences after line 92 in the introduction.
>
> > Indeed, the prime application of the linearity of JL maps is reducing the time complexity of pairwise distance computations [1], which has applications in clustering [2] and nearest neighbor search [3, 7], e.g. for recommender systems and classification. Other examples include compressed sensing [4], which relies on sparse linear measurements to recover an unknown signal. Finally, the linearity of JL has been exploited to speed up least square regression [5] and low-rank matrix approximation [6, 1].
>
> We also remark that in Section 4.4 we provide new results that did not appear in the previous version of the paper. We indeed show that by using a sparse blocked matrix multiplication it is possible to improve the quality of the results with some constraints on the input vectors.
>
> We then kindly ask the reviewer to re-evaluate the paper, in light of this clarification of our results.
>
>  -----
>  References:
>  - [1] Cunningham, John P., and Zoubin Ghahramani. "Linear dimensionality reduction: Survey, insights, and generalizations." The Journal of Machine Learning Research 16.1 (2015): 2859-2900.
>  - [2] Makarychev, Konstantin, Yury Makarychev, and Ilya Razenshteyn. "Performance of Johnson-Lindenstrauss transform for k-means and k-medians clustering." Proceedings of the 51st Annual ACM SIGACT Symposium on Theory of Computing. 2019.
>  - [3] Indyk, Piotr, and Rajeev Motwani. "Approximate nearest neighbors: towards removing the curse of dimensionality." Proceedings of the thirtieth annual ACM symposium on Theory of computing. 1998.
>  - [4] Upadhyay, Jalaj. "Randomness efficient fast-johnson-lindenstrauss transform with applications in differential privacy and compressed sensing." arXiv preprint arXiv:1410.2470 (2014).
>  - [5] Yang, Tianbao, et al. "Fast Sparse Least-Squares Regression with Non-Asymptotic Guarantees." arXiv preprint arXiv:1507.05185 (2015).
>  - [6] Ghojogh, Benyamin, et al. "Johnson-Lindenstrauss lemma, linear and nonlinear random projections, random Fourier features, and random kitchen sinks: Tutorial and survey." arXiv preprint arXiv:2108.04172 (2021).
>  - [7] Ghalib, Abdulaziz, et al. "Clustering and classification to evaluate data reduction via Johnson-Lindenstrauss transform." Future of Information and Communication Conference. Cham: Springer International Publishing, 2020.

---

> > ### Comment · Reviewer_oog7 · 2025-04-04
> >
> > Thank you to the authors for the feedback. I can see now that linearity is a property that is not preserved by my embedding and it is an important property to have in practice. However, it still seems believable that a much simpler analysis could obtain the same bounds that I proposed, but with linearity. In particular, there is existing prior work on analyzing JL type lemmas with complex weights (e.g. see lemma 4 here http://www.cs.cmu.edu/afs/cs/user/dwoodruf/www/knpw11.pdf). I haven't had time to work out the calculations, but lemma 4 in the link is just saying using roots of unity gives us the same moment bounds (and hence concentration) as the standard JL analysis. It is worth thinking if this lemma and the work in that paper (which the authors seem to have missed) already handles much of the technical work of the paper or simplifies the analysis more.

---

> > > ### Author Response · Authors · 2025-04-07
> > >
> > > Thank you for your reply. We are confident that highlighting the linearity of the method, also thanks to your comment, will make our goal clearer, and it will ultimately make for a better paper.
> > >
> > > Thanks also for the new reference, we will add it in the related work and discuss the similarities with our method, namely using complex numbers to to remove bias from estimators. However, this paper has major differences with our approach. In particular, we observed the following:
> > > - The approach of Kane et al. uses a more complicated estimator, that seems to use complex numbers in a fundamentally different way from ours, as they use the $r$-th roots of unity, with $r \in O(\log(\epsilon^{-1}))$. Our techniques use the $4$-th roots of unity, since we have interactions between 4 terms in our estimators (two $x_j$'s and two $w_j$'s), and $\epsilon$ only enters in the dimension of the reduced vectors. Therefore, the role of the roots of unity in the method seems to be different, and the analysis cannot be carried over trivially.
> > > - Moreover, their approach estimates the norm $\lVert x \lVert_p$, but **only** for the range $p \in (0,2)$. Our approach can be extended with small modifications to estimate the $\lVert x \lVert_4$ norm by taking $w =x$. The approach can be extended to estimate the $p$-th norm by taking $p$-th roots of unity, albeit with worse concentration (see also reply to reviewer 2). Therefore, it seems unlikely that the methods by Kane et al. would generalize to weighted norms without major modifications to both the method and the analyis (although this is an interesting research avenue, which we will mention).
> > >
> > > Furthermore, their Lemma 4 doesn't seem to be applicable directly in our case:
> > > - If their Lemma 4 were used to bound our term $\sum_{j_1, j_2, j_3} A_{j_1}A_{j_2}A^2_{j_3} x_{j_1} x_{j_2} w_{j_3}^2$, using $A_{j_1}A_{j_2}A^2_{j_3}$ as their $z_j$ and $x_{j_1} x_{j_2} w_{j_3}^2$ as their $x_j$, it would fail because the $z_j$'s are **not** independent. This is why we resort to Bonami's hypercontractive inequality to bound the tails of the Rademacher chaos, rather than a Khintchine-like inequality. We will remark this in the paper.
> > > - A different application of Lemma 4 could be the following: setting $x'$ (the $x$ of Lemma 4) as our $x \odot w$, would yield that $\mathbb{E}[|\langle x', z\rangle | ^t] \leq f(t) \lVert x \rVert_w^2$. This however doesn't seem to generalize to weighted norms in our setting. Indeed, this approach requires knowing $w$ at the time of the dymensionality reduction, and does not allow to perform multiple queries with different weights.
> > >
> > > We agree with you that a simpler method, or a simpler anlysis of the existing method could exist. However, it is far from trivial to obtain this. We provide some examples of suboptimal approaches below.
> > > - Using existing JL approaches either loses linearity (as discussed before), or requires the weights $w$ to be fixed and known beforehand (as discussed in lines 55-65 of the introduction).
> > > - Using variance-based bounds instead of the Orlicz-norm-based tail bounds (that are one of the technical contribution of our paper) yields and analyisis that is equally complicated, as it involves a Rademacher chaos of order 8. Moreover, we conjecture that it does not yield strong enough concentration to only use averaging in the estimator, and would require using also the median trick, complicating the estimator. We can add a remark about this in the paper.
> > > - Using Hoeffing's inequality to obtain exponentially decaying tail probabilility bounds introduces a dependency on the $L_1$ norm of $x$, which scales linearly with $d$ and yields to suboptimal results. Indeed, the Orlicz-norm-based analysis yields results that seem to be tight in $d$ (see reply to Reviewer 3).
> > >
> > > We hope that we have addressed all your concerns!

---

### Decision · Program_Chairs · 2025-05-01

**Decision:**

Accept (poster)

**Comment:**

This paper presents an embedding of a d-dimensional vector x into k dimension such that, for any d-dimensional weight vector w
 is preserved. Specifically, the authors give an additive error via the classic Johnson-Lindenstrauss guarantee for unweighted vectors, that is, which leads to a multiplicative approximation.

Traditionally, the entries matrices with Johnson-Lindenstrauss moments are subsampled Rademacher random variables. The authors extend this construction by choosing the embedding entries to be complex numbers, where the sign of the complex part is Rademacher distributed. This sketch is applied to input vectors , and is then given to a decoding algorithm. The construction is novel and the use of complex numbers is clever. After this setup, the analysis is not too difficult, but requires controlling stochastic processes in ways that are not standard.

Points to improve: (from the reviews):
it would have been nice if the result had achieved an optimal error, or if the analysis had been tight, or if the authors had presented a framework for generalizing general JL-embedding matrices, rather than only the complex analogue of dense Rademacher sketches. I understand that it might be difficult to improve the paper in the way that I am suggesting.

Alternatively, some experimental focus would have been welcome for a conference like ICML.